# Unraveling the origin of air-stability in single-crystalline layered oxide positive electrode materials

Lei Yu[1,5], Jing Wang[2,5], Tao Zhou [1], Weiyuan Huang [2], Tianyi Li [3], Lu Ma[4], Xianghui Xiao [4], Seoung-Bum Son [2], Steven N. Ehrlich[4], Jianguo Wen [1] ✉, Khalil Amine [2] ✉ & Tongchao Liu [2] ✉

Single-crystalline Ni-rich layered oxides present compelling advantages over conventional polycrystalline counterparts toward large-scale applications, including enhanced mechanical stability and higher energy density. Nevertheless, the deleterious effects of air exposure, which is inevitable in industrial processing, on their structure and electrochemical performance remain poorly understood. Herein, we reveal that air exposure is more detrimental to the electrochemical performance of single-crystalline layered oxide positive electrodes than polycrystalline counterparts. It is found that air-induced surface structural distortions are primarily responsible for the electrochemical performance decay of single-crystalline samples rather than the generally believed surface residual lithium. Leveraging multiscale diffraction and imaging techniques, we identify an undesirable structural transition to a metastable O1* phase, which introduces substantial lattice defects and localized strain concentrations within the layered structure. These adverse structural evolutions compromise structural integrity and promote crack initiation during electrochemical cycling, ultimately accelerating capacity fade. Our findings provide critical insights into the air-induced degradation mechanisms and emphasize the urgent need for developing effective stabilization strategies to facilitate the commercial implementation of single-crystalline Ni-rich positive electrodes.

Surface structure of positive electrode materials plays a crucial role in regulating electron and ion transport dynamics in lithium-ion batteries (LIBs), establishing the material/electrolyte interphase, and ensuring battery cycling stability, especially at the elevated charging voltage for high-energy-density applications[1–3]. This is particularly true for the sought-after nickel-rich layered positive electrode materials (LiNi$_x$Mn$_y$Co$_{1-x-y}$O$_2$, Ni ≥ 80%, NMC). While these Ni-rich NMC materials offer higher energy density and increased lithium utilization, they are susceptible to unstable surface chemistry during electrochemical cycling or air exposure[4–8]. The surface reconstruction is a significant cause of performance degradation during battery operation, and it is closely related to parasitic reactions with the electrolytes[9–12]. Upon charging to high states of charge, the elevated nickel valence increases the surface reactivity, which can trigger a cascade of detrimental side reactions such as electrolyte decomposition, active material dissolution, gas evolution and near-surface phase transitions[13–16]. In

[1]Center for Nanoscale Materials, Argonne National Laboratory, Lemont, IL, USA. [2]Chemical Sciences and Engineering Division, Argonne National Laboratory, Lemont, IL, USA. [3]X-ray Science Division, Advanced Photon Sources, Argonne National Laboratory, Lemont, IL, USA. [4]National Synchrotron Light Source II, Brookhaven National Laboratory, Upton, NY, USA. [5]These authors contributed equally: Lei Yu, Jing Wang. ✉e-mail: jwen@anl.gov; amine@anl.gov; liut@anl.gov

conventional polycrystalline (PC) NMC, these interfacial parasitic reactions are exacerbated due to the inherent anisotropic volumetric changes among primary particles that expose fresh surfaces to the electrolyte[17–21]. In contrast, single-crystalline (SC) NMC significantly enhances mechanical integrity by removing inner grain boundaries and reducing accessible contact area with the electrolyte, thus mitigating performance deterioration during electrochemical cycling[22–24].

On the other hand, surface structural degradation resulting from air exposure represents the most common pre-existing damage for layered positive electrodes before operation[25–29]. Its deterioration mechanism differs markedly from that observed in the electrochemical operation process[30,31]. Specifically, for PC-NMC materials, interactions with ambient $CO_2/H_2O$ primarily occur on the surface of secondary particles, with limited penetration into the interior due to the absence of lattice breathing driven by repeated delithiation/lithiation processes. The continuous reactions at the air/particle interface lead to the generation of residual lithium species and a delithiation layer on the outer layer of the secondary particle, while the inner primary nanograins remain largely unaffected[32–34]. This structural feature of PC layered positive electrode materials highlights the central role of the surface lithium impurities in performance degradation[35]. However, in SC-NMC materials, surface structure becomes increasingly critical as it serves as the sole medium of contact with electrolyte for $Li^+$ extraction/insertion[36,37]. Upon air storage, beyond surface impurity formation, the near-surface structural variations developing throughout the SC particle can substantially influence ion transport, exacerbating the existing kinetic limitations within micron-scale particles[2]. Consequently, air-induced surface structural damage in SC layered positive electrode materials may impose more severe performance penalties than in PC counterparts. Nevertheless, this critical distinction remains insufficiently understood and warrants systematic investigation[38]. Elucidating the air-storage degradation mechanism in SC layered positive electrode materials is thus imperative for further performance optimization and enabling commercial deployment.

The previous study that relied on macroscopic X-ray diffraction and absorption spectroscopy (XRD and XAS) measurements with statistical averaging to evaluate the effects of air exposure on the crystallinity and chemical states of SC Ni-rich layered materials, which is unable to resolve specific microscopic-scale changes and establish a clear link with electrochemical performance[39]. In this work, we conducted multiscale and multidimensional structural characterizations to systematically uncover the air exposure effects on both the structure and electrochemical performance of SC Ni-rich layered positive electrode materials. The results demonstrate that SC-NMC materials suffer from more pronounced performance decay upon air exposure compared to PC systems, where the near-surface lattice distortions play a major role instead of the $Li_2CO_3$-based lithium impurity on the surface. Notably, a novel O1-type delithiation phase (O1*) characterized by a larger d-spacing than the conventional O1 phase emerged in the air-exposed sample. The structural mismatch between the O1* and O3 phases induces significant structural defects and strain concentrations within the otherwise perfect SC lattice. These intrinsic structural degradations not only alter phase transition behavior but also induce mechanical damage during the electrochemical process, leading to rapid capacity decay. This work highlights the critical vulnerability of grain-boundary-free SC layered materials to surface deterioration upon air exposure while emphasizing the necessity of proper storage protocols and surface protection strategies for the development of high-performance SC layered positive electrodes in lithium-ion batteries.

## Results

### Morphology and structure change after air exposure

A representative SC Ni-rich layered positive electrode material, $LiNi_{0.81}Mn_{0.06}Co_{0.13}O_2$ (SC81), was synthesized using a typical co-precipitation method, followed by a calcination step. The pristine SC81 was immediately stored in a glovebox after synthesis, while the SC81-air sample was prepared by exposing the pristine SC81 to ambient atmosphere for 30 days. Energy-dispersive X-ray spectroscopy (EDS) result demonstrates that the atomic ratio of transition metals in the SC81 basically matches the feed ratio, indicating an accurate component control of the co-precipitation synthesis (Supplementary Fig. 1). High-energy XRD was initially employed to elucidate the ensemble-averaged crystal structure of these two samples at the bulk level, as it can detect subtle structural information inaccessible to conventional laboratory XRD. The major XRD reflections of both pristine SC81 and SC81-air samples (Fig. 1a, d) can be well indexed to the hexagonal α-$NaFeO_2$ type crystal structure, corresponding to the $R$-$3m$ space group. The Rietveld refinement analysis of pristine SC81 yields a lattice parameter of $a = b = 2.867$ Å and $c = 14.163$ Å, which is in line with the reported structure for Ni-rich layered positive electrode materials. Compared to pristine SC81, the SC81-air sample undergoes a slight lattice expansion in the $c$ direction and contraction in the $a$ direction, resembling the lattice change observed at the initial delithiation stage during battery charging[40]. This suggests that ambient storage may have led to the oxidation of Ni and the extraction of Li. Moreover, the diffraction patterns of the SC81-air sample show additional weak peaks around a 2θ range of 3–6°, indicating the presence of a $Li_2CO_3$ impurity phase ($C2/c$ space group) with a rough estimate of 1.3% phase fraction.

The molar percentage of lithium residue was further accurately quantified by titration with 0.01 M HCl using an autotitrator (Supplementary Fig. 2). To prevent the formation of additional residual lithium upon contact with air, the residual lithium contents were determined within 10 min of synthesis. The pristine SC81 had a molar ratio of 1.72% for LiOH and 0.55% for $Li_2CO_3$. The slightly high LiOH content may be caused by unreacted excess LiOH during synthesis. After 30 days of air exposure, the $Li_2CO_3$ content largely increased to 3.78%, and only a little LiOH was detected on the surface of the SC81-air sample, suggesting the conversion of LiOH to $Li_2CO_3$ and new $Li_2CO_3$ formation during air exposure. The total molar ratio of Li ions on the surface was 2.82% for pristine samples and 7.60% for air-exposed samples, indicating the extraction of Li ions from the lattice and continuous formation of $Li_2CO_3$ in the presence of air.

The XPS results offer further insights into the surface variations of SC81 upon exposure to air. The C 1$s$ spectra obtained from pristine SC81 (Fig. 1b) mainly contain species originating from adventitious carbon (centered near 284.8 eV), along with a small amount of $(CO_3)^{2-}$ (289.8 eV) and O-C = O (288–289 eV), which well aligns with the titration results. After air exposure, the intensity of the C 1$s$ peak caused by $(CO_3)^{2-}$ increases significantly (Fig. 1e), implying the formation and accumulation of $Li_2CO_3$. In addition, the O 1$s$ peaks arising from $Li_2CO_3$, bulk material, and $OH^-$ are respectively located at 531.7, 528.7, and 530.1 eV (Fig. 1c, f). For the SC81-air sample, the peak of $Li_2CO_3$/LiOH becomes dominant, whereas the M-O peak from the bulk materials becomes less undetectable due to the presence of carbonate species covering the particle surface.

To investigate the impact of air exposure on the chemical state of Ni, Co, and Mn in SC81 materials, synchrotron-based XAS was performed on both pristine SC81 and SC81-air samples. Supplementary Fig. 3 illustrates the X-ray absorption near-edge structure (XANES) spectra of the Mn, Co, and Ni K-edges. No apparent differences were observed in the Mn and Co spectra, but a slight shift towards a higher value was detected in the Ni spectra of the SC81-air sample, indicating an increased oxidation state of Ni. The results of XRD, titration, XPS, and XAS analyses collectively suggest that air exposure caused the extraction of Li ions from the crystal lattice, followed by the oxidation of Ni and the formation of $Li_2CO_3$ on the surface of SC81.

A scanning electron microscope (SEM) was used to detect morphological changes in SC81 following air exposure. The pristine SC81 particles present clean surfaces and fairly uniform particle sizes in the

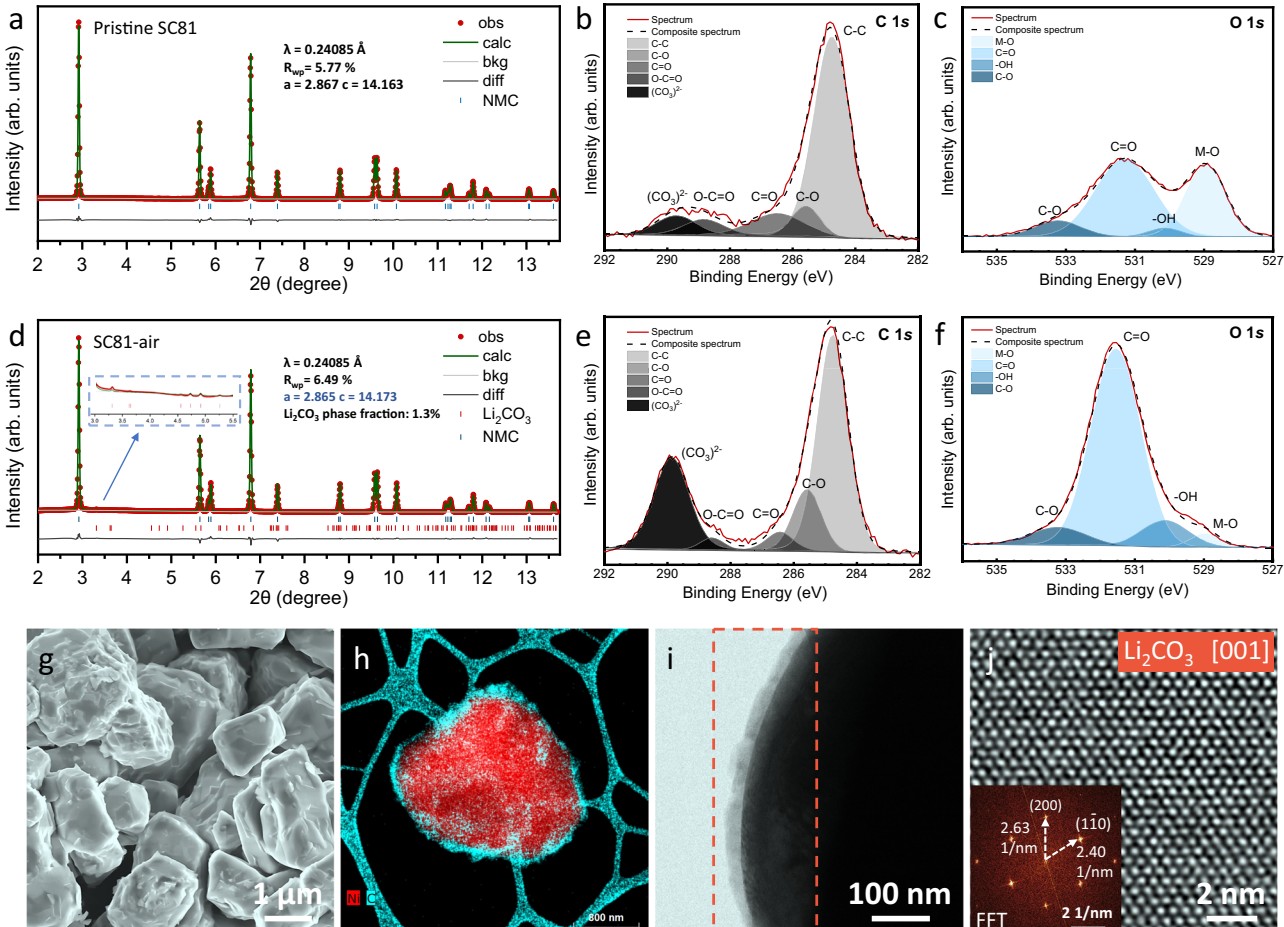

**Fig. 1 | Structure and morphology of SC81-air sample. a, d** XRD Rietveld refinement patterns of SC81-pristine (**a**) and SC81-air sample (**d**). **b, e** XPS C 1s spectra of SC81-pristine (**b**) and SC81-air sample (**e**). **c, f** XPS O 1s spectra of SC81-pristine (**c**) and SC81-air sample (**f**). **g** SEM image of SC81-air. **h** EDS mapping of SC81-air particle. **i** Low-magnification TEM image of the SC81-air surface. **j** Low-dose HRTEM image of the surface impurity and corresponding FFT pattern.

range of 3–5 μm, reflecting well-prepared single-crystalline morphology (Supplementary Fig. 4). After exposure to air, the surfaces of SC81-air particles appeared visibly rough due to the formation of a lithium impurity layer (Fig. 1g). High-angle annular dark-field imaging (HAADF) combined with EDS mapping discloses a carbonaceous layer uniformly enveloping the SC81-air particle surfaces with a thickness of approximately 30 nm (Fig. 1h and Supplementary Fig. 5). The transmission electron microscopy (TEM) image in Fig. 1i highlights the low-contrast and beam-sensitive characteristics of the surface impurity, necessitating a low-dose high-resolution TEM (HRTEM) technique to elucidate its phase structure. As presented in Fig. 1j, the well-preserved atomic lattice displays no noticeable beam damage under this imaging condition. The corresponding fast Fourier transform (FFT) pattern of the HRTEM image aligns well with the $Li_2CO_3$ phase at the [001] zone axis (Supplementary Fig. 6). The uniform $Li_2CO_3$ layer formed across the entire SC particle may vastly increase the resistance for charge transfer due to the electrochemically resistive property of $Li_2CO_3$, thereby influencing the charge/discharge behavior of SC layered positive electrodes.

### Electrochemical performance changes after air exposure

Given the apparent morphological changes following air exposure, it was of particular interest to explore their impact on the electrochemical performance. As illustrated in Fig. 2a, b, the initial discharge capacities within a voltage range of 2.8-4.4 V at a current rate of 0.1 C (20 mA g⁻¹) are recorded as 192.9 and 148.2 mAh g⁻¹ for pristine SC81 and SC81-air, respectively. The observed capacity difference of ~ 23%

between pristine SC81 and SC81-air is considerably larger than the amount of Li ions extracted from the crystal lattice, indicating the change of surface and crystal structure due to ambient storage results in dramatic electrochemical performance deterioration. The initial charging plateau of the SC81-air sample lies at higher voltages with an average increase of 350 mV, suggesting a larger resistance and polarization owing to the presence of substantial $Li_2CO_3$ on the SC81-air surface[28,41,42]. In the subsequent cycles, the overpotential gradually decreases, which can be attributed to the $Li_2CO_3$ decomposition at high voltages. This decomposition is confirmed by EDS and three-dimensional continuous rotation electron diffraction (3D-CRED) measurements, which show the disappearance of $Li_2CO_3$ after 5 cycles (Supplementary Fig. 7, 8). This electrochemical behavior is also reflected on the charge/discharge curves in the voltage range of 2.8–4.5 V (Fig. 2c, d and Supplementary Fig. 9). Figure 2e, f provide the detailed comparisons of the cycling performance of these two positive electrodes operated at a current rate of 0.5 C (100 mA g⁻¹) after three formation cycles at a current rate of 0.1 C (20 mA g⁻¹), and the corresponding coulombic efficiency data is shown in Supplementary Fig. 10. Capacity retentions over 100 cycles were recorded as 86.9% and 83.0% for pristine SC81, and 68.2% and 54.8% for SC81-air, within voltage ranges of 2.8–4.4 V and 2.8–4.5 V, respectively. Clearly, the SC81-air suffers from significant capacity degradation at both voltage ranges, despite a slight capacity recovery after $Li_2CO_3$ was mostly depleted at initial cycles. Therefore, while electrochemically inert $Li_2CO_3$ may affect the initial capacity, it may not be the primary factor contributing to the rapid capacity decay observed

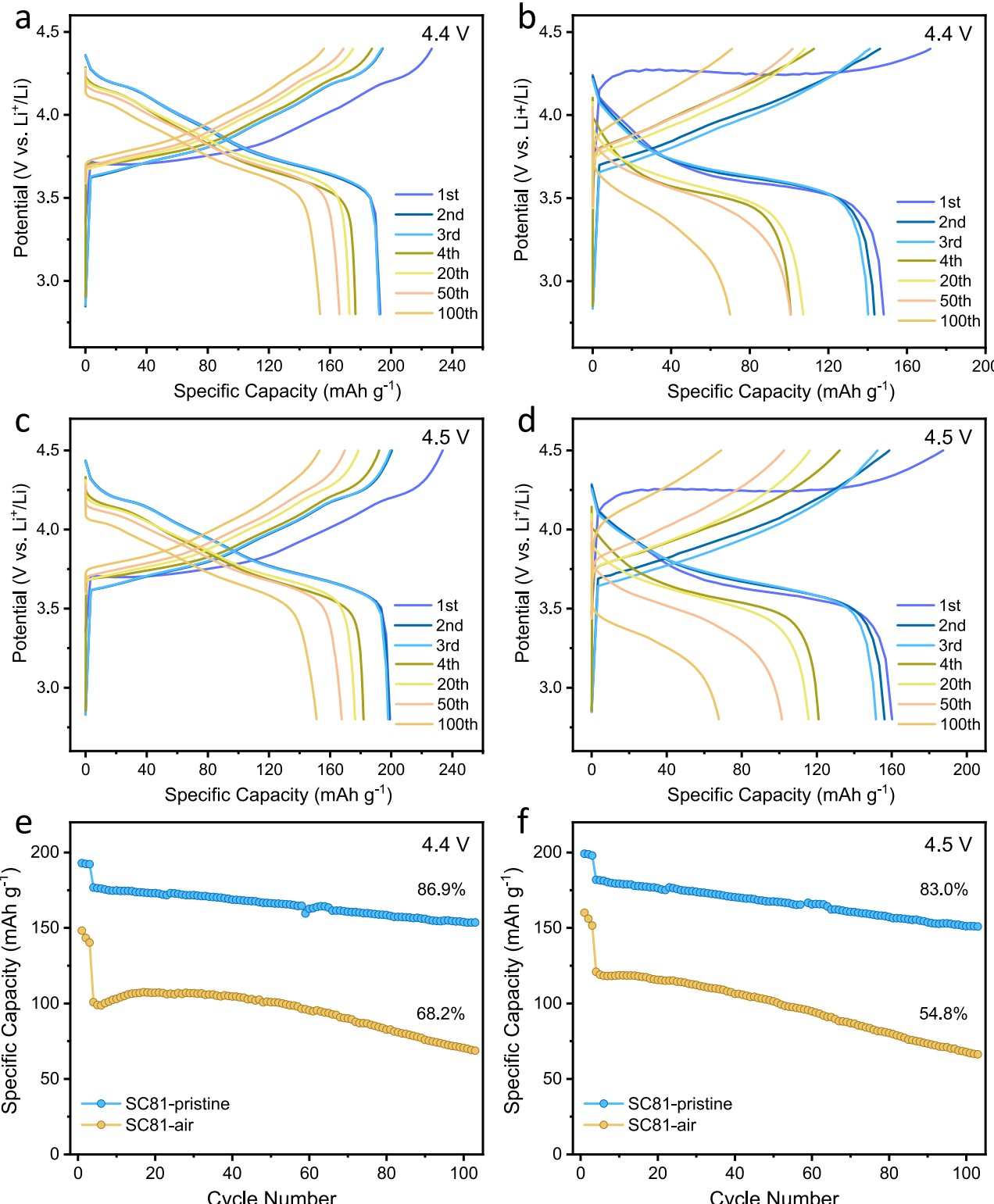

**Fig. 2 | Electrochemical performance comparison between SC81 and SC81-air.** **a**, **b** Charge/discharge curves of pristine SC81 (**a**) and SC81-air (**b**) with voltage ranges of 2.8–4.4 V at 0.1 C (20 mA g$^{-1}$) for first 3 formation cycles, and 0.5 C (100 mA g$^{-1}$) for the rest cycles. **c**, **d** Charge/discharge curves of pristine SC81 (**c**) and SC81-air (**d**) with voltage ranges of 2.8–4.5 V at 0.1 C (20 mA g$^{-1}$) for first 3 formation cycles, and 0.5 C (100 mA g$^{-1}$) for the rest cycles. **e**, **f** Cycle performance of pristine SC81 and SC81-air with a cutoff voltage of 4.4 V (**e**) and 4.5 V (**f**) at 0.1 C (20 mA g$^{-1}$) for first 3 formation cycles, and 0.5 C (100 mA g$^{-1}$) for the rest cycles. These electrochemical tests were performed at a temperature of 25 ± 1 °C.

in the SC81-air sample. To further prove this, the electrochemical performance of pristine SC81 mixed with 3.21% Li$_2$CO$_3$ was tested (Supplementary Fig. 11). The charge/discharge curves show the slight decreases in discharge capacities owing to the introduction of electrochemically inert Li$_2$CO$_3$. The capacity retention after 100 cycles of 84.4% for 2.8–4.4 V is only slightly lower than the pristine SC81, indicating that Li$_2$CO$_3$ did not substantially affect cycling performance.

Furthermore, we conducted the same electrochemical comparisons on the PC layered positive electrode to understand the differences from the SC layered positive electrode (Supplementary Fig. 12). The PC-NMC811-air electrodes still show elevated charging plateaus, but are much lower than those of the SC81-air samples. It is clear that the discharge capacities only have slight decreases compared to the pristine PC-NMC811 samples. Although the PC-NMC811-air electrodes present fast capacity decay, especially at 4.5 V, the capacity retentions are much better than those of the SC81-air samples. These results indicate that PC layered positive electrodes are less susceptible to air-induced degradation compared to SC positive electrodes. SEM and TEM were employed to characterize the morphology and structure of the PC-NMC811-air sample (Supplementary Figs. 13 and 14). As expected, the formation of residual lithium compounds and structure degradation are confined to the outermost surface of the secondary particle, while the inner primary particles remain basically unaffected. These observations indirectly confirm that the more pronounced performance degradation of the SC positive electrode primarily results from the surface structure changes, given that the surface of the SC particle serves as the sole interface exposed to air and electrolyte.

To get insight into the impact of air-induced structural changes on the electrochemical kinetics of SC81 materials, galvanostatic intermittent titration technique (GITT) measurement was conducted to evaluate the chemical diffusion coefficient of Li$^+$ ions ($D_{Li+}$) after 5 cycles when the majority of surface carbonates had been removed. Supplementary Fig. 15 depicts the GITT curves for pristine SC81 and SC81-air samples during the charging process. The calculation results, inserted in Supplementary Fig. 15, reveal that the average $D_{Li+}$ of pristine SC81 during the charging process is $7.68 \times 10^{-12}$ cm$^2$/s, which is approximately 2.9 times higher than that of the SC81-air electrode ($2.68 \times 10^{-12}$ cm$^2$/s). At the initial charging stage, the $D_{Li+}$ values of both samples are relatively low and do not differ much, suggesting a similar near-surface delithiation process. However, as more Li$^+$ ions are mobilized from the inner crystal lattice, the diffusion barrier for Li$^+$ ions in pristine SC81 is significantly reduced, reaching $15.39 \times 10^{-12}$ cm$^2$/s at 3.8 V. In contrast, the $D_{Li+}$ of the SC81-air sample only shows minimal change ($3.94 \times 10^{-12}$ cm$^2$/s at 3.8 V), indicating that the altered surface structure persistently hinders Li$^+$ diffusion throughout the electrochemical process.

## Phase transition mechanism of SC81 after air exposure

To understand the distinct electrochemical performance observed between SC81 and SC81-air samples, in situ HEXRD was performed to track their phase evolution during the first and second cycles (Supplementary Figs. 16 and 18)[40,43]. Figure 3a presents the contour plots of in situ HEXRD data for pristine SC81 between 2.7 V and 4.5 V at a rate of 0.2 C (40 mA g$^{-1}$). Pristine SC81 is characterized by a hexagonal phase H1, which persists during charging below 3.8 V. However, the splitting of (003), (101), (108), and (113) peaks at 3.8 V indicates the emergence of a monoclinic structure (phase M), which progressively replaces phase H1 during the two-phase transformation period (Supplementary Fig. 17). Here, there is an phase separation phenomenon that only appears on the first-cycle delithiation, seemingly deviating from the solid solution reaction. Previous reports reveal that this phase separation is a dynamical artifact and is traced to inter-particle heterogeneity caused by electro-autocatalytic reactions in a many-particle system. As a result, the peak intensities of phase H1 gradually diminish, while the peaks of phase M become increasingly pronounced. At the charging voltage between 3.9 V and 4.2 V, the (003) reflection shifts to lower angles, signaling a gradual expansion of the c-axis and corresponding to the M-H2 phase transformation. This transformation is attributed to the coulombic repulsion between adjacent layers within the delithiated unit cell. As delithiation extends above 4.2 V, a new group of peaks, indicative of phase H3, emerges at higher angles due to a sudden lattice contraction. Upon discharge, the corresponding peaks

exhibit a reverse phase transition process, returning to their original positions. The second electrochemical cycle follows the same sequence of topological reactions observed in the first cycle, except there is no phase separation phenomenon.

Figure 3c depicts the contour plots of the in situ HEXRD data obtained for SC81-air within the voltage range of 2.7–4.5 V. The initial 2θ peak position of the SC81-air is nearly identical to that of pristine SC81 before charging, also corresponding to the hexagonal phase H1. However, the delithiation/lithiation behavior of SC81-air shows notable deviations from that of the pristine SC81. Specifically, the H1-M phase transition begins at 4.3 V, marked by the appearance of a (003) peak at a lower angle and the split of the (101) peak (Supplementary Fig. 19). The elevated onset voltage for the H1-M transition could be ascribed to the presence of Li$_2$CO$_3$ surface coating, which introduces significant overpotential and increased resistance. Subsequently, phase M gradually transitions into phase H2, with a weak peak of phase H3 observable at the end of the charging. Note that the final coexistence of the H$_2$ phase and H$_3$ phase, caused by the sluggish Li diffusion may build severe lattice strain. As presented in Fig. 3b, d, the "active" portion of the SC81-air sample follows a similar phase evolution path as pristine SC81. The capacity reduction of the air-exposed positive electrode should be attributed to the fragile H2-H3 phase transition. During the second cycle, despite a significant decrease in the charging voltage plateau, the phase evolution behavior remains essentially unchanged. Significant broadening of the (003) peak in the SC81-air sample is noted, suggesting a severely delayed phase transition stays in the following cycle. This illustrates that the surface structure changes significantly affect the bulk electrochemical behavior. Although no drastic H2-H3 transition occurs, the delayed phase transition and the resulting lattice strain will exacerbate the capacity decay of SC81-air during long-term cycling.

## Detailed structural characterization of SC81 after air exposure

The aforementioned 3D-CRED test also manifests the structural information of the air-exposed particle (Supplementary Fig. 8). Obvious streak diffraction along the c* direction is identified, implying the presence of lattice distortion and stacking faults within the layered structure[44,45]. To delve into the detailed structural changes of the sample itself post-air exposure, HRTEM was used to characterize the surface structure of the SC81-air sample. As shown in Fig. 4a and Supplementary Fig. 20, it is clear that the surface layered structure becomes considerably disordered due to the uneven extraction of lithium. The selected area electron diffraction (SAED) pattern in Fig. 4b reveals two distinct sets of diffraction spots in the crystal structure. Apart from the conventional O3 layered phase, an O1-type phase has been found and labeled. Normally, the O1 phase (AB-AB-AB oxygen stacking sequence) is formed by interlayer gliding of O3-type layered positive electrode materials at the high delithiation stage (H$_2$-H$_3$ transition), with a concomitant reduction in interlayer spacing[46,47]. However, in our case, the newly observed O1-type phase resulting from air-exposed delithiation differs significantly from the traditional O1 phase produced by high charging voltages. The diffraction spot distance of this O1-type phase along the c* direction is shorter than that of the O3 phase, suggesting a larger interlayer spacing in real space. Hence, this phase is denoted as O1* to distinguish it from the conventional O1 phase.

Enlarged views of two distinct areas marked in Fig. 4a further illustrate the coexistence of O1* and O3 phases (Fig. 4c, d). The direct differences between these phases in TEM imaging can be identified by the stacking of TM atoms along the [001] direction: the O3 structure exhibits an αβγ-stacking pattern, whereas the O1* phase displays an ααα-stacking pattern[48]. Consequently, the lattice distortion is attributed to the lattice parameter misfit between the coexisting O1* and O3 phases. The enlarged view in Fig. 4e reveals the presence of an edge dislocation defect at the lattice distortion interface. This defect is more

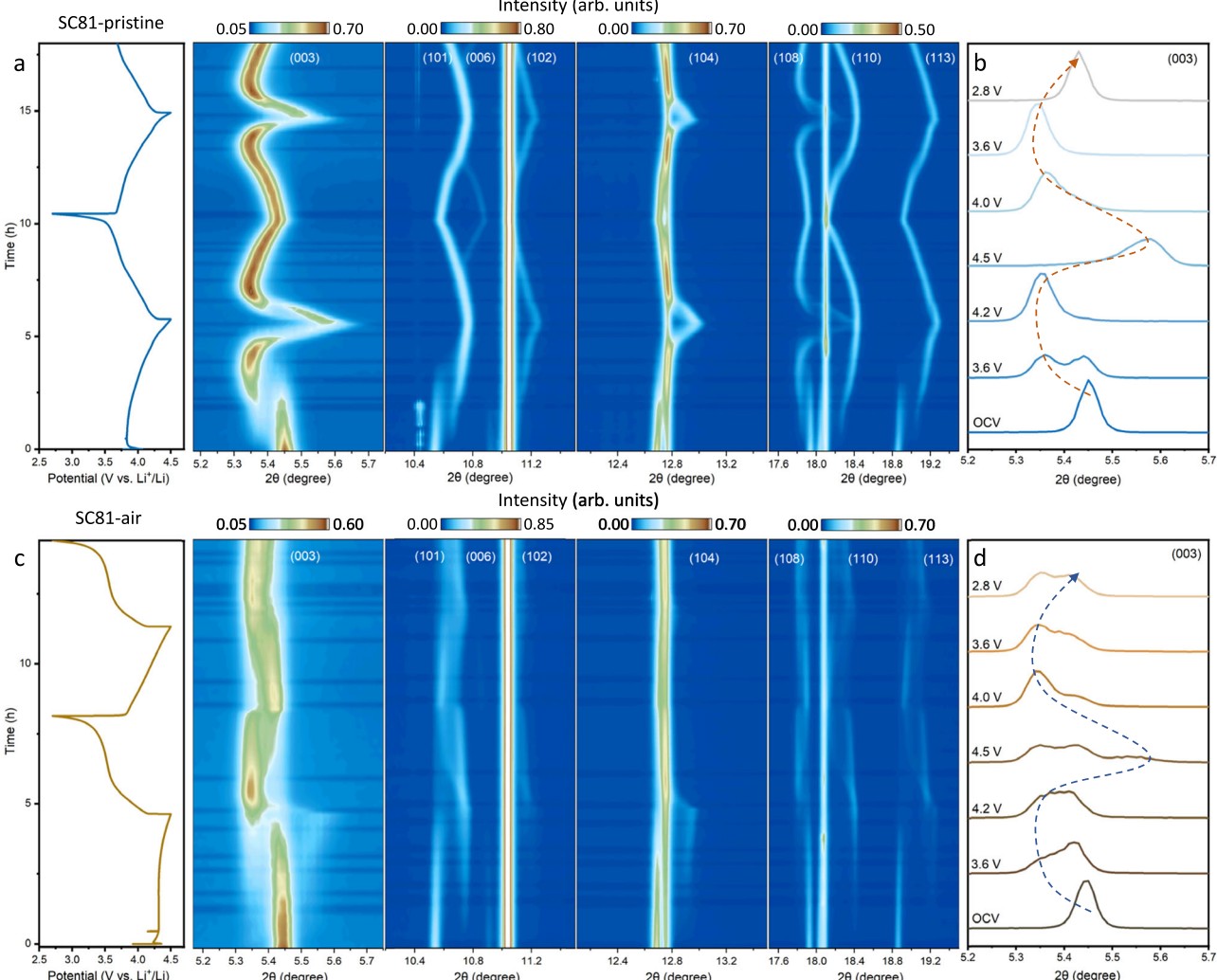

**Fig. 3 | In situ HE-XRD characterization. a, c** Enlarged in operando synchrotron XRD contour plots and corresponding charging/discharging curves of pristine SC81 (**a**) and SC81-air (**c**) with crystal planes denotations. The batteries underwent two charge-discharge cycles at a specific current of 0.2 C (40 mA g⁻¹) and a temperature of 25 ± 1 °C within the voltage ranges of 2.7−4.5 V. **b, d** The selected (003) reflection of pristine SC81 (**b**) and SC81-air (**d**) at different voltages during the first cycle. The arrow lines indicate the reaction direction and "active" phase evolution pathway.

clearly observed in the inverse FFT image which only masks the interlayer diffraction spots (Fig. 4f). Geometric phase analysis (GPA) results further demonstrate that these structural irregularities induce significant lattice strain concentrated in this region (Fig. 4g). Detailed TEM analysis in Supplementary Fig. 21 shows the distribution of the O1* phase. Similar to the conventional O1 phase, the increased energy barriers and reduced diffusion sites in O1* are thought to impede Li⁺ transport, as evidenced by the GITT results. Overall, the air-exposed sample shows the additional introduction of the O1* phase on the particle surface, a phenomenon not previously reported. The lattice mismatch between O1* and O3 phases leads to severe stacking faults, structural dislocations, and lattice strain concentration, which may be the root cause of the electrochemical performance degradation observed after air exposure.

Full-field transmission X-ray microscopy (TXM) combined with three-dimensional (3D) XANES was conducted to revel the chemical state variations in the particles. Figure 5a–c presents uniform color distribution in both the 3D TXM-XANES mapping and its cross-sectional two-dimensional (2D) mappings, indicating a well-distributed Ni-related phase and a homogeneous Ni oxidation state in pristine SC81. In comparison, the observed color in the SC81-air particle surface turns red, indicating the increased valence state of Ni with Li₂CO₃ impurity formation after air exposure (Fig. 5e–g). The statistical analysis of the whiteline peak position for each 3D dataset further certifies the slightly increased valence state of Ni in SC81-air, as evidenced by the shift towards higher energy (Fig. 5d, h). The inhomogeneous chemical state distribution of Ni related to the varying lithium content is likely to induce localized structural change and lattice distortion.

The lattice strain at the particle level was investigated using synchrotron X-ray nano-probe technique[49–51]. Figure 5i shows a high-resolution Ni fluorescence map on the pristine SC81 sample. Because of their random orientations, each nanoparticle diffracts at different sample and detector angles, only one of them is visible in the integrated intensity map (Fig. 5j) acquired at the same time. Figure 5k, i show the Ni fluorescence and the integrated intensity map of a SC81-air particle. To ensure a fair comparison, all the selected particles are not only similar in size and shape but also have their (003) planes aligned in the same direction. Standard tilt-series analysis yields a 2D projected image of the 3D nanoparticle, which tends to under-estimate the strain values as they are averaged along the projected direction. To uncover the actual strain of the oxidized surface layer, we first reduce the 5D dataset to a 1D rocking curve. As is shown in Fig. 5m, while the rocking

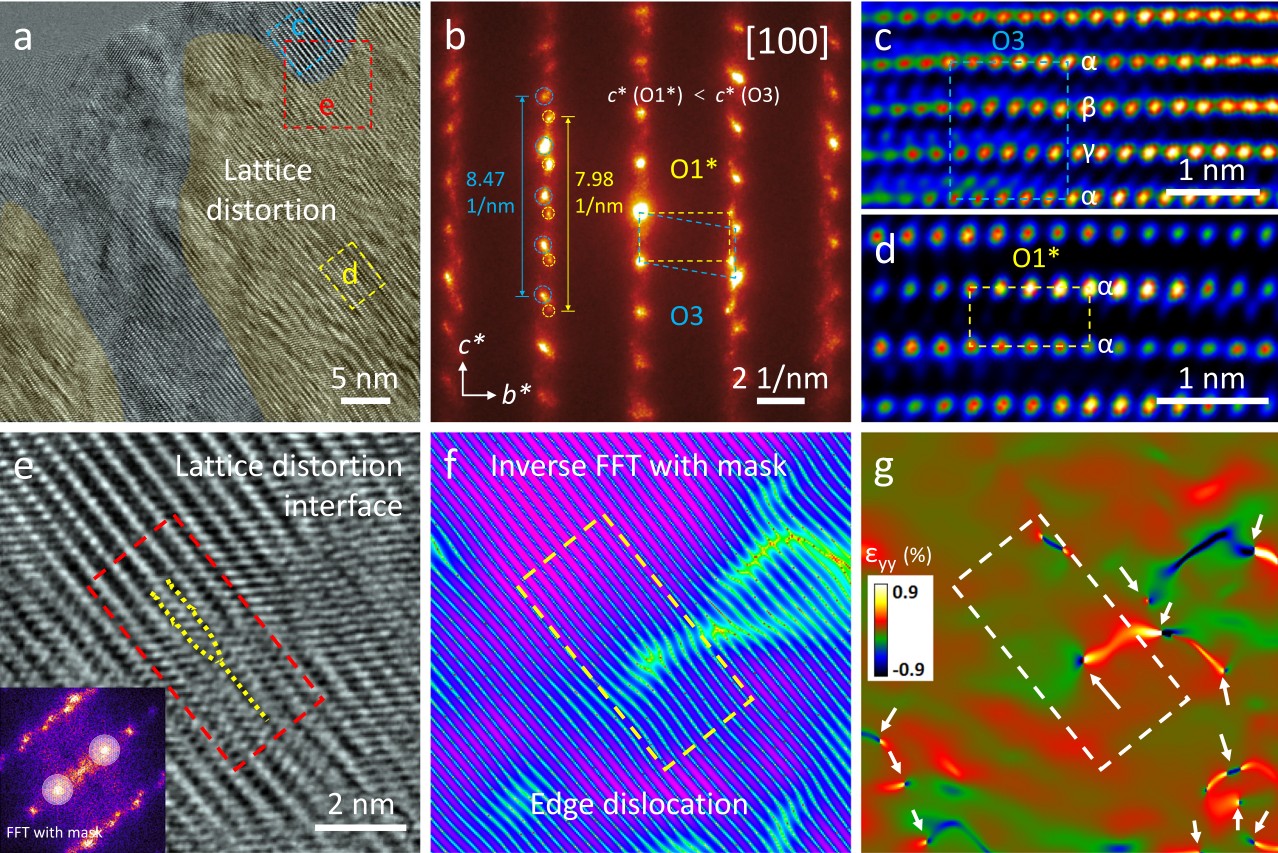

**Fig. 4 | High-resolution TEM characterization. a** HRTEM image of SC81-air after ion-milling showing the surface structure. **b** SAED pattern acquired at the position of (**a**). **c**, **d** The enlarged views of the areas marked with blue and yellow rectangles in (**a**). **e** The enlarged views of the areas marked with red rectangle in (**a**), the inset is corresponding FFT pattern. **f** Inverse FFT image applying mask, the mask is shown in the inset of (**e**). **g** Corresponding GPA result of the area of (**e**).

curve of the pristine particle shows a symmetric peak shape around the (003) reflection, that of the air-exposed particle shows an additional broad peak, centered around $2q \sim 14.7°$, corresponding to a tensile strain of ~ 3%. The lattice change resulting from this additional peak is consistent with the d-spacing change of the O1* phase detected by HRTEM. Next, dark field maps are generated to localize the origin of the tensile strain. When using intensities scattered within the unstrained $2q$ range, a rather uniform contrast was observed in the dark field image shown in Fig. 5n, as is expected for a particle before cycling. When using intensities scattered within the $2q$ range corresponding to the large tensile strain, the left and right edges of the particle lit up (Fig. 5o). This confirms that the areas with the large tensile strain (~ 3%) are indeed located at the surface of the air-exposed particle. We note that tensile strain was only observed on the side surfaces due to their larger cross-section with the incident beam. The observed value is a more accurate determination of the strain state of the oxidized surface layer than large beam XRD analysis. The latter reports a strain of only ~ 0.07% as the value is averaged over the entirety of the largely unstrained particles.

### Structural characterization after electrochemical cycles

Given the evident structural changes and disparities in electrochemical performance, the detailed structures of SC81 and SC81-air samples after 100 cycles were further detected using TEM. No obvious morphological damage is observed in the cycled SC81 sample (Supplementary Fig. 22). The HRTEM image shows only conventional rock salt phase transition on the surface due to electrolyte attack, while the interior of the particle maintains a well-defined layered structure. In contrast, the cycled SC81-air sample exhibits a different structural

degradation mechanism, with severe particle damage observed (Fig. 6a and Supplementary Fig. 23). Several intragranular cracks aligned with the layered structure appear, which is the result of internal strain release via mechanical degradation. The HRTEM image and corresponding FFT pattern (Fig. 6b, c) on the crack demonstrate irreversible rock salt phase transition, and line-scan electron energy loss spectroscopy (EELS) (Fig. 6d) shows a reduction in Ni valence at the crack[52]. This suggests that the intragranular cracks facilitate electrolyte penetration, triggering undesirable side reactions inside the SC particles. The rock salt phase is typically electrochemically inactive, leading to the significant capacity decrease observed in the electrochemical tests. HRTEM imaging (Fig. 6e, f) reveals the lattice distortion propagates into the bulk structure of the cycled SC81-air with cycling, further incubating the inner crack and worsening the electrochemical performance[53,54]. These TEM results indicate that the surface structural changes, including O1* phase formation, lattice distortion, structural dislocation, and microstrain accumulation after air exposure, are responsible for the rapid capacity decline.

### Fundamental of O1* and structural degradation mechanism under air exposure

Supplementary Fig. 24 provides a schematic representation of the O3, O1, and O1* phases. The formation of the O1-type phase (AB-AB-AB oxygen stacking sequence) typically originates from interlayer gliding of the O3-type layered structure during delithiation. However, the driving forces behind the formation of O1 and O1* phases differ. The conventional O1 phase forms through deep delithiation at high voltages. As Li+ ions are extracted, their roles as electrostatic shielding effect and structure "pillars" between oxygen layers weaken, reducing

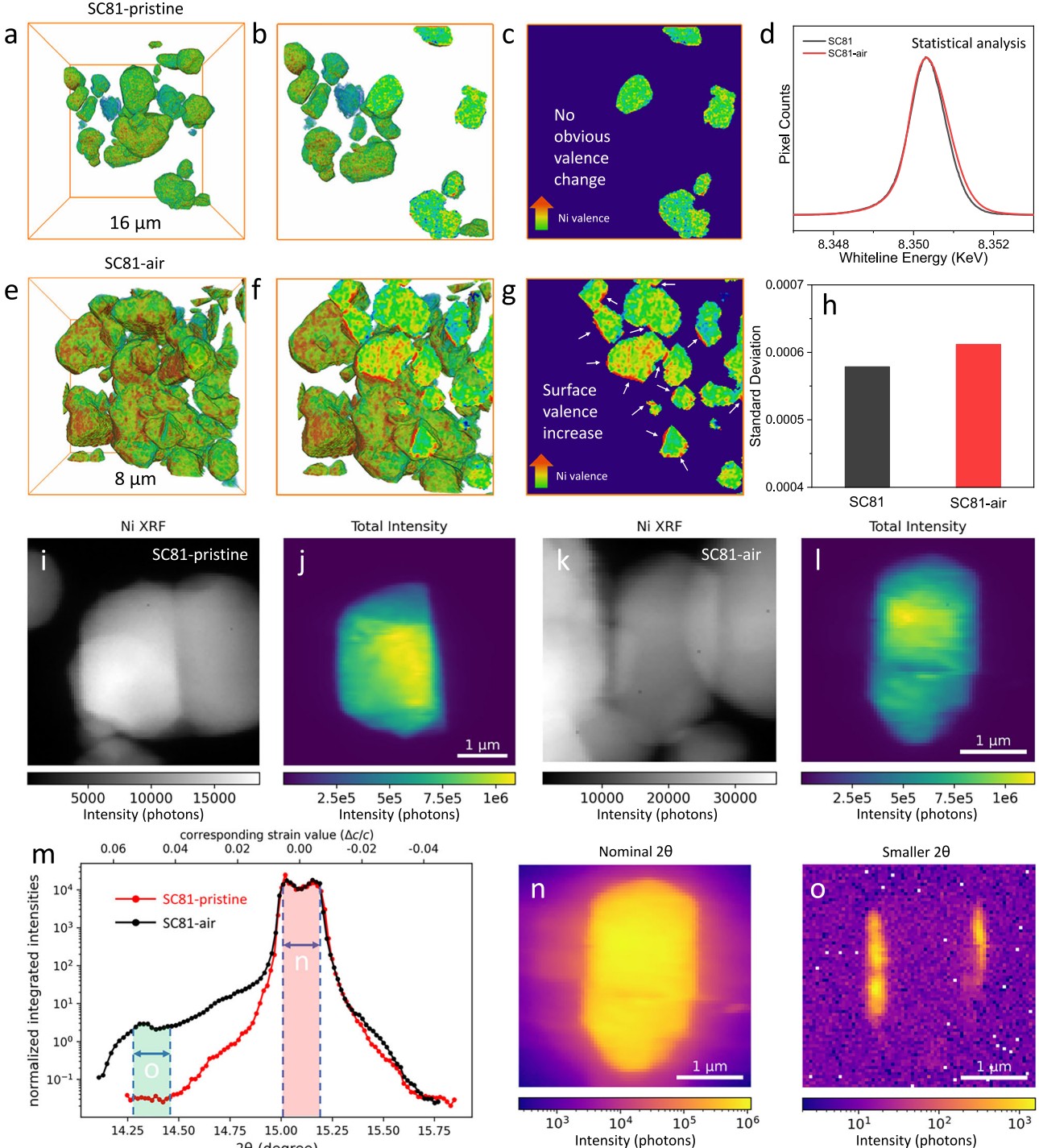

**Fig. 5 | 3D TXM-XANES and SXDM for chemical state and strain distribution.**
**a**–**c** The 3D TXM-XANES mapping (**a**) and cross-sectional views (**b**, c) of SC81-prisitne. **d** Statistic distribution based on whiteline peak position. **e**–**g** The 3D TXM-XANES mapping (**e**) and cross-sectional views (**f**, **g**) of SC81-air. **h** The calculated standard deviation of the whiteline peak position in statistic based on (d). **i, j** XRF image (**i**) and SXDM image (**j**) of SC81-pristine. **k, l** XRF image (**k**) and SXDM image (**l**) of SC81-air. **m** The strain distribution profile. **n, o** The corresponding integration of unstrained (**n**) and strained parts (**o**), respectively.

structural support. Simultaneously, the heterogeneous reaction caused by sluggish Li diffusion induces shear-strain, which eventually triggers gliding along the (003) plane in the fragile structural framework, facilitating the transformation from ABC to AB oxygen stacking. This rearrangement helps relieve lattice stress/strain and reduce interlayer electrostatic repulsion. Moreover, at high voltages, the increased covalency of transition metal (TM)−O bonds initiates ligand-to-metal charge transfer, leading to the delocalization of electron density from oxygen to the transition metal (e.g., Ni). This process reduces the localized negative charge on oxygen and further mitigates interlayer repulsion. Consequently, at high delithiation states, the combined effects of oxygen charge delocalization and the inherently denser packing of AB-stacked oxygen layers contribute to the contraction of interlayer spacing in the O1 phase. In contrast, air exposure

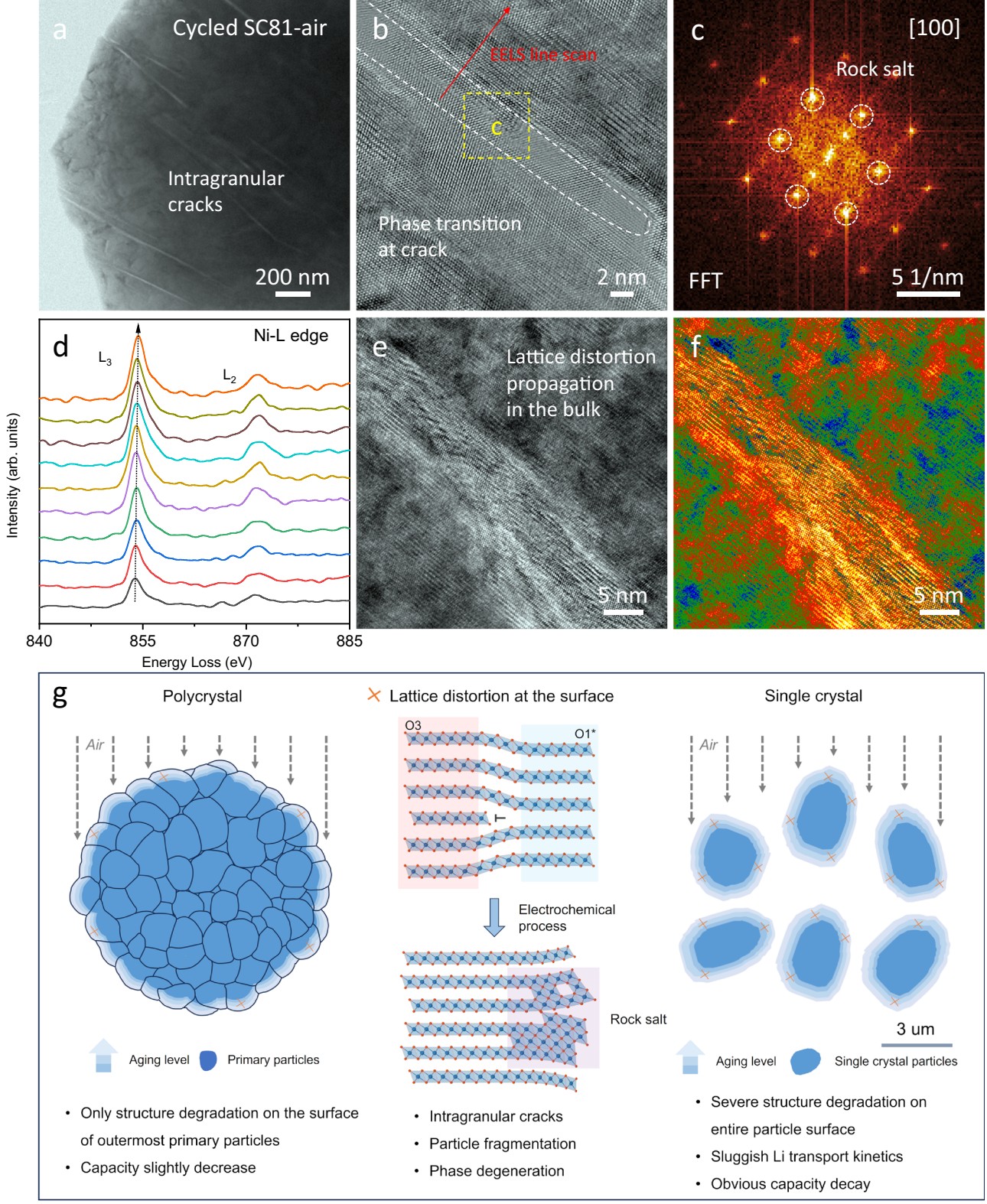

**Fig. 6 | Microscopic TEM observation for cycled SC81-air and structural degradation mechanism under air exposure. a** Low-magnification TEM image of discharged SC81-air sample after 100 cycles at a specific current of 0.5 C (100 mA g⁻¹) and a temperature of 25 ± 1 °C within the voltage ranges of 2.8–4.5 V. **b** HRTEM image of the crack. **c** Corresponding FFT pattern. **d** Line-scan EELS along the arrow direction in (**b**). **e, f** HRTEM image (**e**) and color HRTEM images (**f**) in the bulk. **g** The comparison between PC and SC positive electrodes and the corresponding structural degradation mechanism under air exposure. The blue spheres in structural models denote Ni atoms, and the red spheres denote O atoms.

triggers uneven Li$^+$ extraction from the particle surface, causing localized lattice strain accumulation rather than deep delithiation states. This inhomogeneous process induces partial interlayer gliding transitions from O3 to O1* stacking, particularly concentrated near the surface area, as directly evidenced by SXDM and HRTEM measurements. Crucially, the limited lithium removal under chemical delithiation weakens the local shielding effect while amplifying interlayer electrostatic repulsion. These competing factors result in the expanded interlayer spacing characteristic of the O1* phase. Moreover, our experiments indicate that the O1* phase is unstable and tends to transform into the rock salt phase under external stimuli. As shown in Supplementary Fig. 25, the O1* phase disappears and transforms into the rock salt phase after electron beam irradiation. This instability accelerates structural degradation towards the rock salt phase during electrochemical cycling in air-exposed samples.

Figure 6g presents a schematic diagram illustrating the structural degradation mechanism under air exposure, which highlights the substantial differences between SC and PC layered positive electrode materials. In terms of PC positive electrodes, their agglomerated spherical secondary particle morphology effectively protects the internal primary particles from air-induced damage. The formation of Li$_2$CO$_3$ impurity and structural degradation are confined to the outermost surface of secondary particles, resulting in slight decreases in electrochemical capacity and stability. On the contrary, the surface of SC positive electrode particles is completely exposed. A long-time air exposure will lead to forming a Li$_2$CO$_3$-based lithium impurity coating over the entire particle surface, accompanied by lithium extraction from the surface lattice. However, the Li$_2$CO$_3$ layer is found to be fully decomposed after formation cycles, and fast capacity decay remains during prolonged battery cycling. Therefore, we propose that surface structural degradation of SC-NMC upon air exposure plays an undeniable role in electrochemical decay. Specifically, the extraction of lithium from the surface lattice induces a transition of the surface O1* phase, which results in significant lattice distortion, structural dislocation, and lattice strain accumulation in the layered structure. These structural degradations are shown to constrain the H$_2$-H$_3$ phase transition, hinder Li ion diffusion, and exacerbate reaction heterogeneity and lattice contradictions during electrochemical processes, ultimately resulting in crack formation, structural breakdown, and rapid capacity fading. Overall, the air aging process undermines the structural stability advantage that SC layered positive electrode materials typically boast. This insight can inspire the research in chemical process engineering, particularly in developing advanced protective coatings (such as Al$_2$O$_3$, LiPO$_x$ or phosphate ester coatings) to prevent structural degradation of SC layered positive electrode materials upon air exposure[55]. These pre-set protective coatings are considered to be more promising than post-exposure washing and laborious re-calcination, as mentioned in some reports.

## Discussion

In summary, SC Ni-rich layered positive electrode material undergoes significant surface degradation upon exposure to air. The surface lithium impurity coating formed at this process greatly impedes the diffusion of Li$^+$ ions but is wholly consumed within the first few cycles. Surface lattice changes, specifically involving an O1* phase transition, have been identified as the primary cause of electrochemical performance decay. This phase transition introduces substantial structural defects and microstrains into the pristine SC lattice, compromising the mechanical integrity and creating pathways for further surface side reactions during the electrochemical processes. As a result, this ongoing structural instability propagates throughout the SC particles, leading to a marked decline in specific capacity and cyclic stability. This work demonstrates that SC layered positive electrodes completely lose their structural advantages in electrochemical applications following long-time air exposure. Therefore, proper protection of SC

Ni-rich layered positive electrode materials during storage and transportation is critical, as merely washing away surface lithium impurities may not be sufficient as a reliable modification strategy.

## Methods

### Materials synthesis

The pristine SC81 was synthesized via a coprecipitation reaction followed by a high-temperature sintering procedure. First, the precursor was produced via the hydroxide coprecipitation in a 4 L continuous stirring tank reactor (CSTR), maintaining pH at 12 and temperature at 55 °C. In the reaction process, the average particle size of the precipitate was controlled at ∼ 5 μm. The resulting precipitate was aged overnight, washed with deionized water, and vacuum-dried at 80 °C for 12 h. Next, the obtained precursor was thoroughly blended with LiOH·H$_2$O (Sigma-Aldrich, ≥99%) at a Li/TM molar ratio of 1.03. The mixture was calcined in a tube furnace under an oxygen atmosphere, first at 920 °C for 2 h and then held at 850 °C for 12 h. Finally, the collected powder was ground and sieved through a 400-mesh sieve to obtain the pristine SC81. The SC81-air sample was prepared by exposing the pristine SC81 in air for 30 days. The temperature for air storage is approximately 25 °C, and the relative humidity is around 43%.

### Titration

The residual lithium content in each sample was determined with an acid-base titration. A certain amount of sample powder was put into 30 mL of deionized water for 10 min and then filtered to collect a clear solution. Afterward, the filtrate was titrated with 0.01 M HCl (Easy pH, Mettler Toledo) down to pH 3, and the residual lithium contents as well as individual LiOH and Li$_2$CO$_3$ contents were determined by 2-equivalence points. The residual lithium content of the pristine sample was determined within 10 min after preparation to minimize the formation of residual lithium in contact with air.

### Electrochemistry tests

The samples were fabricated into positive electrodes through the slurry-casting method to evaluate the electrochemical performance. Active materials, carbon black (C45 Conductive Carbon Black, TIMCAL), and polyvinylidene fluoride (PVDF, Solvay® 5130 PVDF binder) at a mass ratio of 8:1:1 were mixed in n-methyl-2-pyrrolidone (NMP) solvent to form a uniform slurry. The prepared slurry was cast on Al foil (20 μm thick) and dried at 80 °C in a vacuum for 12 h. After that, the dried laminate was punched into 14 mm positive electrode disks with an active material mass loading of ∼ 4 mg cm$^{-2}$. The obtained positive electrode together with a Li negative electrode (200 μm thick), a Celgard 2325 separator (PP/Polyethylene, PE/PP, thickness: 25 μm, porosity: 39%, average pore size: 0.025 μm, lateral dimension: 100 mm), and 40 μL of electrolyte (1 M LiPF$_6$ in EC/EMC; 3/7 by volume, H$_2$O content below 20 ppm) was sealed into 2032-type coin cells in a glovebox (H$_2$O and O$_2$ below 0.1 ppm). The coin cells were cycled at various current rates in a designated potential window with a galvanostatic charged-discharged protocol in a Maccor electrochemical analyzer at 25 ± 1 °C in ambient air. The C-rate is defined as 1 C = 200 mA g$^{-1}$. Unless specified, 3 cycles of the formation process were conducted at 0.1 C (20 mA g$^{-1}$) before the test. GITT measurements were conducted in the versatile multichannel potentiostat system during the charging process. The cells were set to relax for 2 h after every 20 min at discharging/charging rates of 0.1 C (20 mA g$^{-1}$).

### Synchrotron X-ray measurements

The crystal structures of positive electrode materials were characterized using HEXRD on sector 17-BM of the Advanced Photon Source (APS) at Argonne National Laboratory. A high-energy X-ray beam (λ = 0.24085 Å) was employed in transmission geometry to acquire two-dimensional diffraction patterns. These patterns were recorded

with a Perkin-Elmer large-area detector positioned 700 mm from the samples. Subsequent Rietveld refinement of the HEXRD data was conducted using the GSAS software package.

In situ time-resolved HEXRD measurements during electrochemical cycling were also performed at APS 17BM with a wavelength of 0.45202 Å and a detector placed 400 mm from the coin cells. Synchrotron HEXRD offers high penetration and low absorption characteristics, enabling precise, real-time monitoring of bulk structural evolution under realistic operating conditions. This is particularly beneficial for detecting subtle phase changes often obscured in conventional lab-scale XRD due to poor signal-to-noise ratio and limited temporal resolution. The 2032-type coin cells were customized with a 3 mm hole to allow X-rays to pass through, and diffraction patterns were collected every 10 min.

XANES spectroscopy at the Ni K edge, Mn K edge and Co K edge was carried out on beamline 7-BM of National Synchrotron Light Source II (NSLS-II) at Brookhaven National Laboratory. Incident X-ray photon energy was monochromatized using a Si(111) double-crystal monochromator. To eliminate higher-order harmonic contributions, the monochromator crystals were detuned, attenuating the incident beam intensity by ~30%. All the spectra were acquired in the transmission mode at room temperature.

### 3D-Continuous rotation electron diffraction

3D-CRED data acquisition was conducted on the FEI Talos F200X TEM. For TEM sample preparation, the sample particles were first ultrasonically dispersed in ethanol and then dropped onto a lacey carbon TEM grid. Subsequently, the prepared grid was loaded into the TEM column using a Fischione Model 2550 Cryo Transfer Tomography Holder. During the 3D-CRED test, the particle of interest was positioned at the eccentric height and started with a continuous single tilt from −45° to 45° at a rate of $0.6° s^{-1}$. Concurrently, diffraction patterns were recorded as a video stream (100 ms per frame) utilizing the Velox software. The acquired video data was converted into the MRC format for subsequent processing with the *REDprocessing software package*[56].

### Scanning electron microscopy and transmission electron microscopy

SEM morphology observation was conducted using JEOL JSM-7100F. TEM, EDS, SAED, EELS, and HRTEM characterization were carried out using the Argonne chromatic aberration-corrected TEM (ACAT) (a FEI Titan 80–300ST with an image aberration corrector to compensate for both spherical and chromatic aberrations) at an accelerating voltage of 200 kV. For the TEM observation, the samples were thinned using a commercial Gatan precision ion polishing system (PIPS) to prepare TEM specimens. To prepare cycled samples for TEM, the cycled coin cells were disassembled within an Ar-filled glove box at 25 ± 1 °C. The retrieved positive electrodes were promptly rinsed with dimethyl carbonate and then dried completely under vacuum. The dried samples were subsequently thinned using the same PIPS procedure as the pristine sample.

### Scanning X-ray diffraction microscopy

Scanning X-ray diffraction and fluorescence experiments were conducted on the CNM-APS 26-ID-C hard X-ray nanoprobe beamline at Argonne National Laboratory. For strain analysis, a 10 keV incident beam, focused using a 160 μm Fresnel Zone Plate with an outermost zone width of 30 nm, was raster-scanned over a $4 \times 4 \, \mu m^2$ sample area with a 50 nm spatial resolution under high vacuum to prevent oxidation. Diffraction patterns were collected using an Eiger 2 X 1 M detector at a distance of 1 m from the sample. The raster scan was repeated at 16 different rocking curve angles spanned over 3°, in a process known as k-mapping or tilt series[57]. Because a two-dimensional diffraction pattern is collected at each point on the area detector, this process is equivalent to acquiring a three-dimensional reciprocal space map about the (003) reflection, separately for each scanned point on the nanoparticle. All the dark field images presented in the work are results from the reduction of this 5-dimensional (3 dimensions in the reciprocal space + 2 dimensions in the real space) dataset. X-ray fluorescence (XRF) signal was also collected during the raster scans using a Vortex ME-7 detector. The resulting XRF maps were used primarily for aligning the 16 diffraction maps using image registration methods.

### Full-field transmission X-ray microscopy imaging

3D nano-XANES datasets were acquired using TXM imaging on the beamline 18-ID FXI of NSLS-II at Brookhaven National Laboratory. At each energy point, the projection images taken from 0 to 180° were processed to reconstruct 3D tomography. The voxel resolution of X-ray microscopy imaging was 40 nm.

## Data availability

Source data are provided in this paper.

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

## Acknowledgements

This work gratefully acknowledges support from the U. S. Department of Energy (DOE), Office of Energy Efficiency and Renewable Energy, Vehicle Technologies Office. Argonne National Laboratory is operated

for the DOE Office of Science by UChicago Argonne, LLC, under contract number DE-AC02-06CH11357. Work performed at the Center for Nanoscale Materials, a U.S. Department of Energy Office of Science User Facility, was supported by the U.S. DOE, Office of Basic Energy Sciences, under Contract No. DE-AC02-06CH11357. This research used resources of the Advanced Photon Source (17-BM, 11-ID-C and 26-ID-C), a U.S. Department of Energy (DOE) Office of Science User Facility operated for the DOE Office of Science by Argonne National Laboratory under Contract No. DE-AC02-06CH11357. Use of the National Synchrotron Light Source II (beamline 7-BM and 18 ID) is supported by the US Department of Energy, an Office of Science user Facility operated by Brookhaven National Laboratory under contract number DE-SC0012704.

## Author contributions

L.Y., T. Liu and K.A. conceived of and designed the experiments. J. Wang and T. Liu synthesized all the materials and conducted electrochemical measurements. L.Y. and J. Wen carried out the TEM, 3D-CRED and EELS tests. T.Z. conducted SXDM experiments and analysis. J. Wang, T. Li, L.M., and S.N.E. performed ex situ/in situ synchrotron HEXRD and XAS. W.H. and X.X. performed TXM and data analysis. S.-B.S. conducted XPS experiments and analysis. L.Y., J. Wang, J. Wen, T. Liu and K.A. wrote the manuscript and all authors edited the manuscript.

## Competing interests

The authors declare no competing interests.
