## [Peer Review File · Nature Communications]

Unraveling the origin of air-stability in single-crystalline layered oxide positive electrode materials

Corresponding Author: Dr Tongchao Liu

Version 0:

Reviewer comments:

Reviewer #1

(Remarks to the Author)

The manuscript from Yu et al. presented interesting failure mechanism of single crystal high nickel NMC cathode after air exposure. The authors attribute the decreased capacity and capacity retention of air exposed SC Ni81 to a novel O1* phase formation due to partially delithiation due to H+/Li+ exchange. This manuscript would benefit from a more nuanced discussion, with appropriate further in-depth discussions of the mechanism of O1* phase change, the driving force behind this, and why this is fundamentally different from electrochemical de-lithiation. Given the lack of any other significant failure modes, I would consider this manuscript potentially valuable. A few comments follow on the content, data interpretations and opinions enunciated by the authors. I recommend the authors to consider the following points:

1. Please provide the details for the air exposure conditions, e.g., temperature, relative humidity.
2. Please provide EDS characterization of the SC81-pristine sample as in Figure S5 for air exposed sample. In Figure S5, the EDS results suggest, that I quote, "The atomic ratio of C:transition metal (TM) is calculated to be 6.56%, nearly equivalent to molar ratio of Li extraction in titration test". If the EDS results align with the titration, shouldn't the C:TM ratio be 6.56/2 % instead given the titration result of Li:TM of 6.45% in figure S2 (the chemical formula of Li2CO3).
3. I am very concerned about the quality of the electrochemical data, specially the V-Q curves of pristine and air exposed Ni81 as shown in Figure 2, are this result representative? Any duplicate results?
4. To further rule out the impact of Li2CO3 from the conclusion, I would encourage the authors try new experiments with adding similar amount of Li2CO3 into the pristine SC Ni81 to test the electrochemical result and cycling.
5. It might be helpful to further understand the in situ XRD experiment results better if the author could plot the expanded view of plots showing the changes of Li2CO3 in Figure S12.
6. In figure 3, the C rate should be 0.2C instead of 0.1C as the authors claimed in the manuscript.
7. Please explain why the phase transition is only observed for the initial charge, not the followed discharge in in-situ XRD.
8. O1* is directly related to H+/Li+ exchange during air exposure, if Li2CO3 (or LiOH) can be observed from the XRD pattern, why this O1* phase is not present in the XRD pattern?
9. What is reason fundamentally for forming this O1* phase instead of O3 (electrochemical de-lithiation) or O1 (deep delithiation)?
10. The authors claim the difference of failure mechanisms between SC and PC high Ni material with air exposure, however there is no experimental backed results to support this conclusion, this can only be the authors' presumption, please either provide concrete experiments and in-depth analysis, or make extensive revision to the manuscript.

Reviewer #2

(Remarks to the Author)

The Manuscript entitled "Unraveling the origin of air-stability in single-crystalline layered cathodes" by Yu et al. describes a study of Ni-rich NMC materials degradation due to air exposure of the as synthesized materials. The study by itself is very sound, uses relevant techniques to support the authors' main statements, and allows for some interesting speculations about dominating degradation mechanisms. To this end, the authors compare single crystals and polycrystalline cathode materials, proving a different degradation behaviour.

While the study is well described, using extensive high-level characterization, it is not clear to me how easy it would be to exclude air exposure over the complete manufacturing process and how much their conclusions are addressing a real problem in larger scale chemical engineering. The reactivity of eventually formed components in an uncontrolled synthesis

process is clear to have an impact on the electrochemical cycling of the cathodes made from these materials. This maybe an important point to mention and is not necessarily a weak point of the manuscript. It could even further supply ideas in chemical process engineering, especially in the field of protective coatings. This may lift the paper to the level required for a multidisciplinary journal as Nature Communications. Concerning the scanning hard X-ray diffraction tool, the authors may cite recent applications of this method to single crystalline cathodes during cycling, further underlying the importance of such advanced characterization methods for the understanding of complex structural transformations inside battery cathode crystals during cycling like e.g. published by Martens et al. (<https://doi.org/10.1038/s41467-023-42285-4>).

Reviewer #3

(Remarks to the Author)

This manuscript reveals the impact of air exposure for single-crystalline nickel-rich layered cathode materials. The authors emphasized near-surface lattice distortions play a major role in the decay of single-crystalline Ni-rich cathodes instead of the Li₂CO₃-based lithium impurity on the surface. However, many reports have demonstrated that air-induced surface degradation led to the decay of Ni-rich cathodes. The findings in this work are not impressive enough to be published in Nature Communications, such a high-level journal.

1. The authors claim that the newly formed O1* phase, which has a higher Ni valence, is key to the deterioration of electrochemical performance. This finding is intriguing and novel. However, the paper seems somewhat lacking in detailed exploration of the mechanism. The detailed structure of the O1* phase and the origin of its formation are unclear. The main text notes that the new O1* phase differs significantly from the traditional O1 phase produced by high charging voltages, as it has a larger d-spacing. The reason for the formation of a new O1-type phase should be further investigated. Is the O1* phase more stable, or does it have a different Li content from the general O1 phase, etc.?
2. The authors assert that surface side reactions between the surface and air in single-crystalline samples are more pronounced than in polycrystalline ones, but no polycrystalline data were provided, including electrochemical performance and characterizations, which is unconvincing. The authors should provide data on polycrystalline samples, preferably using PC samples with a similar specific area as the SC sample, to determine if the exposed surface area is the critical factor influencing performance.
3. The core point that has been emphasized in this work is that "surface lattice distortions" has a greater impact on SC NCM than surface residual lithium does on SC NCM. However, the evidence is not enough. Many characterizations in the manuscript confirmed the bad effect of surface residual lithium. It is confusing. Experimental settings need to be adjusted to prove the conclusions of the paper.
4. Some comparisons do not make much sense, such as Fig. 2d, e. First, air exposed SC81 is definitely worse than unexposed SC81. Second, surface residual lithium induced impedance increase or rock salt phase can also lead to higher capacity loss than the amount of Li ions extracted from the crystal lattice.
5. The degradation of Ni-rich cathode in air is a multi-factor process. Only several STEM or TEM images at local region is not enough to rule out other factors in the decay of SC81.
6. The schematic diagram in Fig. 6g is too similar to a recently published Science paper Fig. 5 (<https://www.science.org/doi/10.1126/science.ado1675>). Proper revisions should be made to avoid any similarities.
7. The performance and capacity are significantly inferior to previously published work of single-crystalline Ni-rich cathodes in Nature Communications. Such as Nature communications, 2021, 12(1), 5320.

Version 1:

Reviewer comments:

Reviewer #1

(Remarks to the Author)

Thanks for providing the comments. All my questions are addressed, so this work can be accepted by the Journal of Nature Communications.

Reviewer #3

(Remarks to the Author)

In the revised manuscript, the author responded to my concerns in detail and convincingly. Additional electrochemical data, SEM and STEM images of PC-NCM811 cathode with/without air exposure were provided. Furthermore, 3D-CRED results of SC81-air particles before and after 5 cycles, and the electrochemical data of pristine SC81 mixed with Li₂CO₃ also addressed my concern. New data further confirmed the authors' conclusion. In my opinion, this version is now proper to be published in Nature Communications.

Itemized Responses/Revisions to the original manuscript

Article Reference: NCOMMS-24-33182

Manuscript Title: “*Insight into structural degradation of single-crystalline nickel-rich layered cathode under air exposure*”

Responses to reviewer 1:

General comment: “The manuscript from Yu et al. presented interesting failure mechanism of single crystal high nickel NMC cathode after air exposure. The authors attribute the decreased capacity and capacity retention of air exposed SC Ni81 to a novel O1 phase formation due to partially delithiation due to H⁺/Li⁺ exchange. This manuscript would benefit from a more nuanced discussion, with appropriate further in-depth discussions of the mechanism of O1* phase change, the driving force behind this, and why this is fundamentally different from electrochemical de-lithiation. Given the lack of any other significant failure modes, I would consider this manuscript potentially valuable. A few comments follow on the content, data interpretations and opinions enunciated by the authors.”*

Response: We would like to thank the reviewer for his/her positive feedback and valuable suggestions, which definitely help us make this work more solid and appealing. Based on the reviewer’s suggestions, we have carefully revised this manuscript. Below is point-by-point response to the comments.

1. Comment #1

Comment: “Please provide the details for the air exposure conditions, e.g., temperature, relative humidity.”

Response: Thank you for your suggestion. The experimental conditions for air storage are detailed in the methods section of the revised manuscript. The temperature is approximately 25 °C, and the relative humidity is around 43%.

2. Comment #2

Comment: “Please provide EDS characterization of the SC81-pristine sample as in Figure S5 for air exposed sample. In Figure S5, the EDS results suggest, that I quote, “The

atomic ratio of C:transition metal (TM) is calculated to be 6.56%, nearly equivalent to molar ratio of Li extraction in titration test”. If the EDS results align with the titration, shouldn't the C:TM ratio be 6.56/2 % instead given the titration result of Li:TM of 6.45% in figure S2 (the chemical formular of Li_2CO_3).”

Response:

Thank you for your question, which brought to our attention an incorrect description in the manuscript. The EDS-calculated C:transition metal (TM) atomic ratio of 6.56% is inconsistent with the results of chemical titration. Chemical titration is a highly accurate quantitative method for measuring lithium residues in cathode materials, and its result, a Li:TM ratio of 6.45%, is considered reliable. Although EDS is also a dependable technique for elemental analysis, we overlooked the influence of the surrounding carbon support film on the calculation of elemental carbon content.

As shown in Figure R1, the EDS-derived C:TM ratio for pristine SC81 is 2.46%, which is significantly higher than the Li_2CO_3 content determined by titration. This discrepancy suggests that the carbon support film on the TEM grid introduces a significant artifact in the measurement of carbon content. Therefore, using the EDS-derived C:TM ratio to evaluate lithium extraction is not appropriate. In response, we have removed the corresponding statement from the revised manuscript.

Figure R1. EDS characterization of the pristine SC81 particle.

3. Comment #3

Comment: “I am very concerned about the quality of the electrochemical data, specially the V-Q curves of pristine and air exposed Ni81 as shown in Figure 2, are this result representative? Any duplicate results?”

Response: Thanks for your comments. We further supplied two sets of charge/discharge curves of SC81-air within voltage ranges of 2.8–4.5 V in the revised manuscript (Fig. 2c-d) and revised supporting information (Supplementary Fig. 9). As shown in Figures R2 and R3, all the charge/discharge curves of SC81-air electrodes show the similar electrochemical behavior, with an initially increasing charging plateau that decreases in subsequent cycles as well as rapid capacity decay. These collective results indicate that the electrochemical performance of single-crystal cathode deteriorates dramatically after air exposure.

Figure R2. a, b, Charge/discharge curves of pristine SC81 (a) and SC81-air (b) with voltage ranges of 2.8–4.4 V at 0.1 C for first 3 formation cycles, and 0.5 C for the rest cycles. c, d, Charge/discharge curves of pristine SC81 (c) and SC81-air (d) with voltage ranges of 2.8–4.5 V at 0.1 C for first 3 formation cycles, and 0.5 C for the rest cycles. e, f, Cycle performance of pristine SC81 and SC81-air with a cutoff voltage of (e) 4.4 V and (f) 4.5 V at 0.1 C for first 3 formation cycles, and 0.5 C for the rest cycles.

Figure R3. Another set of charge/discharge and cyclic performance curves of pristine SC81 and SC81-air with voltage ranges of 2.8–4.5 V at 0.1 C for first 3 formation cycles, and 0.5 C for the rest cycles.

4. Comment #4

Comment: “To further role out the impact of Li_2CO_3 from the conclusion, I would encourage the authors try new experiments with adding similar amount of Li_2CO_3 into the pristine SC Ni81 to test the electrochemical result and cycling.”

Response: Thanks for your advice. We have further tested the electrochemical performance of pristine SC81 mixed with 3.21% Li_2CO_3 (the detected Li_2CO_3 content of SC81-air in the titration test). As shown in Figure R4, the charge/discharge curves show slight decreases in discharge capacities owing to the introduction of Li_2CO_3 . The capacity retentions after 100 cycles are 84.4% for 2.8–4.4 V and 82.0% for 2.8–4.5 V, which are also only slightly lower than that of the pristine SC81 (revised manuscript Fig. 2). These results indicate that Li_2CO_3 by itself does not seriously deteriorate electrochemical performance.

Figure R4. The electrochemical performance plots of pristine SC81 mixed with 3.21% Li_2CO_3 .

5. Comment #5

Comment: “It might be helpful to further understand the in situ XRD experiment results better if the author could plot the expanded view of plots showing the changes of Li_2CO_3 in Figure S12”

Response: Yes, it would be very helpful if we could show the in situ XRD results regarding changes in Li_2CO_3 . However, the formation of Li_2CO_3 is a minor surface phase transformation that occurs upon exposure to air. In the synchrotron XRD analysis of the powder samples, only small characteristic peaks related to Li_2CO_3 are observed in the magnified pattern (Fig. 1, revised manuscript). When using assembled cells for in situ XRD tests, the presence of the binder, current collector, and Kapton film significantly increases the background noise (Figure R5), which makes surface Li_2CO_3 undetectable. It is noteworthy that a diffraction peak at approximately 6.3° appears to correspond to the position of Li_2CO_3 characteristic peak. However, it can be concluded that this is not the case, as this peak is present in both pristine and air-exposed samples, regardless of their electrochemical states. Moreover, diffraction peaks at this position are also observed in previously reported XRD patterns of fresh electrode samples^[1]. Therefore, we believe this peak likely corresponds to the diffraction of some component within the battery, rather than Li_2CO_3 .

[1] Liu, T. et al. Rational design of mechanically robust Ni-rich cathode materials via concentration gradient strategy. *Nat. Commun.* 12, 6024 (2021).

Figure R5. Some selected XRD curves from the in-situ XRD dataset.

6. Comment #6

Comment: “In figure 3, the C rate should be 0.2C instead of 0.1C as the authors claimed in the manuscript.”

Response: Thanks very much for your reminder. We have corrected this typo in the revised manuscript.

7. Comment #7

Comment: “Please explain why the phase transition is only observed for the initial charge, not the followed discharge in in-situ XRD.”

Response: Thanks for your comment. It is noted that a phase separation phenomenon between H1 and H2 is observed during initial charging. This phenomenon has been reported in numerous operando XRD studies^[2,3] and has even been described as a “first-cycle effect”. It is actually a dynamic artifact related to inter-particle heterogeneity caused by electro-autocatalytic reactions in a many-particle system. We have added an explanation of this phenomenon in the revised manuscript.

[2] Park, J. et al. Fictitious phase separation in Li layered oxides driven by electro-autocatalysis. *Nat. Mater.* 20, 991–999 (2021).

[3] Lee, W. et al. New insight into Ni-rich layered structure for next-generation Li rechargeable batteries. *Adv. Energy Mater.* 8, 1701788 (2017).

8. Comment #8

Comment: “O1* is directly related to H⁺/Li⁺ exchange during air exposure, if Li₂CO₃ (or LiOH) can be observed from the XRD pattern, why this O1* phase is not present in the XRD pattern?”

Response: Thanks for your question. As mentioned above, the formation of Li₂CO₃ is only a minor surface phase transformation that occurs upon exposure to air. The formation of the O1* phase is also like this. According to the previous report^[4], the characteristic peaks of the O1 and O3 phases largely overlap and are only shown as broadening of specific characteristic peaks [(10l) and (01l)]. Not to mention the fact that this is only a minor phase transition (using synchrotron XRD can find the small characteristic peaks of Li₂CO₃ in the magnified pattern), so we cannot distinguish the O1* phase from the XRD pattern.

[4] Croguennec, L., Pouillier, C. & Delmas C. Structural characterisation of new metastable NiO₂ phases. *Solid State Ion.* 135, 259–266 (2000)

9. Comment #9

Comment: “What is reason fundamentally for forming this O1* phase instead of O3 (electrochemical de-lithiation) or O1 (deep delithiation)?”

Response: Thank you for your question. Figure R6 provides a schematic representation of the O3, O1, and O1* phases. The formation of the O1-type phase (AB-AB-AB oxygen stacking sequence) typically originates from interlayer gliding of the O3-type layered structure during delithiation^[5]. However, the driving forces behind the formation of O1 and O1* phases differ. The conventional O1 phase forms through deep delithiation at high voltages. As Li⁺ ions are extracted, their roles as

electrostatic shielding effect and structure “pillars” between oxygen layers weaken, reducing structural support. Simultaneously, the heterogeneous reaction caused by sluggish Li diffusion induces shear-strain, which eventually triggers gliding along the (003) plane in fragile structural framework, facilitating the transformation from ABC to AB stacking. This rearrangement helps relieve lattice stress/strain and reduce interlayer electrostatic repulsion. Moreover, at high voltages, the increased covalency of transition metal (TM)–O bonds initiates ligand-to-metal charge transfer, leading to the delocalization of electron density from oxygen to the transition metal (e.g., Ni). This process reduces the localized negative charge on oxygen and further mitigates interlayer repulsion. Consequently, at high delithiation states, the combined effects of oxygen charge delocalization and the inherently denser packing of AB-stacked oxygen layers contribute to the contraction of interlayer spacing in the O1 phase.

In contrast, air exposure triggers uneven Li^+ extraction from particle surface, causing localized lattice strain accumulation rather than deep delithiation states. This inhomogeneous process induces partial interlayer gliding transitions from O3 to O1* stacking particularly concentrated near the surface area, as directly evidenced by SXDM and HRTEM measurements. Crucially, the limited lithium removal under chemical delithiation weakens the local shielding effect while amplifying interlayer electrostatic repulsion. These competing factors result in the expanded interlayer spacing characteristic of the O1* phase.

We have added a paragraph with this description in the revised manuscript.

[5] Li, S. et al. Sustainable LiCoO_2 by collective glide of CoO_6 slabs upon charge/discharge. *Proc. Natl. Acad. Sci. USA*. 119, e2120060119 (2022).

Figure R6. Schematic diagram of the O3, O1, and O1* phases.

10. Comment #10

Comment: “The authors claim the difference of failure mechanisms between SC and PC high Ni material with air exposure, however there is no experimental backed results to support this conclusion, this can only be the authors’ presumption, please either provide concrete experiments and in-depth analysis, or make extensive revision to the manuscript.”

Response: Thank you for your suggestion. we conducted the same electrochemical comparisons on the PC cathode to understand the differences from the SC cathode

(Figure R7). The PC-NMC811-air electrodes still show elevated charging plateaus but are much lower than those of the SC81-air samples. It is clear that the discharge capacities only have slight decreases compared to the pristine PC-NMC811 samples. Although the PC-NMC811-air electrodes present fast capacity decay, especially at 4.5 V, the capacity retentions are much better than those of the SC81-air samples. These results indicate that PC layered cathodes are less susceptible to air-induced degradation compared to SC cathodes.

SEM and TEM were employed to characterize the morphology and structure of the PC-NMC811-air sample (Figures R8 and R9). As expected, the formation of residual lithium compounds and structure degradation are confined to the outermost surface of the secondary particle, while the surface of inner primary particles remain basically unaffected. These observations strongly support our conclusion in this work. For PC layered cathodes, despite the destruction of the outermost surface, the grain boundaries still serve as Li^+ diffusion channels for the intact internal primary particles, leading to minimal performance degradation. In contrast, for SC layered cathodes, the particle surface is the sole medium for Li^+ extraction and insertion, and its breakdown results in significant performance failure.

Figure R7. a, b, Charge/discharge curves of pristine PC-NMC811 (a) and PC-NMC811-air (b) with voltage ranges of 2.8–4.4 V at 0.1 C for first 3 formation cycles, and 0.5 C for the rest cycles. c, d, Charge/discharge curves of pristine PC-

NMC811 (c) and PC-NMC811-air (d) with voltage ranges of 2.8–4.5 V at 0.1 C for first 3 formation cycles, and 0.5 C for the rest cycles. e, f, Cycle performance of pristine PC-NMC811 and PC-NMC811-air with a cutoff voltage of (e) 4.4 V and (f) 4.5 V at 0.1 C for first 3 formation cycles, and 0.5 C for the rest cycles.

Figure R8. The SEM observations of the PC-NMC811-air particles, showing the residual lithium compounds on the surface of secondary particle.

Figure R9. The SEM and STEM images of the crashed PC-NMC811-air particles. The inner primary particles show clean surfaces and no structural degradation.

Responses to reviewer 2:

General comment: “The Manuscript entitled “Unraveling the origin of air-stability in single-crystalline layered cathodes” by Yu et al. describes a study of Ni-rich NMC materials degradation due to air exposure of the as synthesized materials. The study by itself is very sound, uses relevant techniques to support the authors’ main statements, and allows for some interesting speculations about dominating degradation mechanisms. To this end, the authors compare single crystals and polycrystalline cathode materials, proving a different degradation behaviour.

While the study is well described, using extensive high-level characterization, it is not clear to me how easy it would be to exclude air exposure over the complete manufacturing process and how much their conclusions are addressing a real problem in larger scale chemical engineering. The reactivity of eventually formed components in an uncontrolled synthesis process is clear to have an impact on the electrochemical cycling of the cathodes made from these materials. This maybe an important point to mention and is not necessarily a weak point of the manuscript. It could even further supply ideas in chemical process engineering, especially in the field of protective coatings. This may lift the paper to the level required for a multidisciplinary journal as Nature Communications. Concerning the scanning hard X-ray diffraction tool, the authors may cite recent applications of this method to single crystalline cathodes during cycling, further underlying the importance of such advanced characterization methods for the understanding of complex structural transformations inside battery cathode crystals during cycling like e.g. published by Martens et al. (<https://doi.org/10.1038/s41467-023-42285-4>).”

Response: Thank you so much for the positive comments on the significance of this work and the valuable suggestions. We believe these suggestions significantly improve the quality of the article. We responded to the comments point by point as follows.

Actually, we do not need to take elaborate measures to exclude air exposure during the manufacturing process. Layered cathode materials are typically prepared by calcining the precursor in an O₂ atmosphere. After preparation, these materials are immediately transferred to a glove box, but always exposed to air for a few minutes. However, air aging of layered cathode materials is a slow and gradual process. The few minutes of exposure after synthesis are negligible compared to long-term air exposure during ambient storage.

In recent years, single-crystal layered transition metal oxides have gained significant attention as next-generation high-performance cathode materials of lithium-ion battery. Their storage and handling during commercial production are practical challenges for large-scale applications. Our work addresses a thorough study of the air stability of single-crystal layered cathode materials, revealing their structural decay mechanism and the impacts on electrochemical performance. Single-crystal layered cathode materials are usually prepared by adding more additional lithium in high-temperature calcination. These samples contain higher amounts of surface lithium residues, and their surface serves as the only lithium-ion diffusion channel. As a result, the single-crystal layered cathode materials show more severe air-aging issue compared to their polycrystalline counterparts.

Thanks again for your insightful comments and for pointing out the potential relevance of our findings to broader chemical engineering contexts. We appreciate your recognition of the significance of our work in understanding the degradation of layered structure during air exposure. In response to your comment on how our conclusions address real problems in large-scale chemical engineering, we would like to clarify that our findings highlight the importance of controlling surface structure stability in single crystal Ni-rich layered oxides. Specifically, we demonstrate that the degradation of the surface layered structure plays a more crucial role in electrochemical performance decay than the formation of Li_2CO_3 and LiOH . This insight can inspire the research in chemical process engineering, particularly in developing advanced protective coatings (such as Al_2O_3 , LiPO_x or phosphate ester coatings) to prevent structural degradation and enhance air stability. We argue that pre-set protective coatings are more promising than post-exposure washing and laborious re-calcination, as mentioned in some reports. We have revised the manuscript to emphasize this point and included a discussion on how our findings could contribute to the field of protective coatings, broadening the practical implications of our work to a multidisciplinary audience.

We also appreciate your suggestion to cite recent work on scanning hard X-ray diffraction by Martens et al. This reference provides a significant insight into the structural dynamics of single-crystalline cathodes during cycling, which further supports the importance of our methodology in characterizing complex structural transformations. We believe that including this citation strengthens our discussion on the relevance of advanced characterization methods for battery materials.

Responses to reviewer 3:

General comment: "This manuscript reveals the impact of air exposure for single-crystalline nickel-rich layered cathode materials. The authors emphasized near-surface lattice distortions play a major role in the decay of single-crystalline Ni-rich cathodes instead of the Li_2CO_3 -based lithium impurity on the surface. However, many reports have demonstrated that air-induced surface degradation led to the decay of Ni-rich cathodes. The findings in this work are not impressive enough to be published in Nature Communications, such a high-level journal."

Response: Thank you for your feedback and for taking the time to evaluate our manuscript. While many reports have addressed the air-aging issue of polycrystalline Ni-rich cathodes, single-crystal layered oxides, as a newly morphology-designed cathode material, have not been extensively studied in this context. In recent years, single-crystal layered cathodes have garnered significant attention from both academia and industry, particularly with the ongoing efforts toward large-scale commercial production. As a result, the storage and handling of single-crystal materials have become critical considerations.

In this study, we present a comprehensive investigation into the structural and electrochemical performance of single-crystal Ni-rich cathodes following air exposure. Specifically, we reveal the microscopic changes in the surface structure induced by air exposure using multiscale and multi-dimensional characterization techniques, and propose an O1-type phase transition mechanism, which has not been reported previously. We emphasize the far-reaching effects of these microstructural changes on both structural and performance degradation. Our findings demonstrate that the surface instability of single-crystal materials is more detrimental to performance than that observed in polycrystalline counterparts. This work provides valuable insights into the storage and protection of single-crystal materials, which are key challenges for their practical application.

We appreciate your insights and comments, which have helped us refine and address the potential limitations in our work. Below, we provide a point-by-point response to your comments.

11. Comment #1

Comment: *"The authors claim that the newly formed O1* phase, which has a higher Ni valence, is key to the deterioration of electrochemical performance. This finding is intriguing and novel. However, the paper seems somewhat lacking in detailed exploration of the mechanism. The detailed structure of the O1* phase and the origin of its formation are unclear. The main text notes that the new O1* phase differs significantly from the traditional O1 phase produced by high charging voltages, as it has a larger d-spacing. The reason for the formation of a new O1-type phase should be further investigated. Is the O1* phase more stable, or does it have a different Li content from the general O1 phase, etc.?"*

Response: Thanks for your insightful question. Figure R10 provides a schematic representation of the O3, O1, and O1* phases. The formation of the O1-type phase

(AB-AB-AB oxygen stacking sequence) typically originates from interlayer gliding of the O3-type layered structure during delithiation^[5] However, the driving forces behind the formation of O1 and O1* phases differ. The conventional O1 phase forms through deep delithiation at high voltages. As Li^+ ions are extracted, their roles as electrostatic shielding effect and structure “pillars” between oxygen layers weaken, reducing structural support. Simultaneously, the heterogeneous reaction caused by sluggish Li diffusion induces shear-strain, which eventually triggers gliding along the (003) plane in fragile structural framework, facilitating the transformation from ABC to AB stacking. This rearrangement helps relieve lattice stress/strain and reduce interlayer electrostatic repulsion. Moreover, at high voltages, the increased covalency of transition metal (TM)–O bonds initiates ligand-to-metal charge transfer, leading to the delocalization of electron density from oxygen to the transition metal (e.g., Ni). This process reduces the localized negative charge on oxygen and further mitigates interlayer repulsion. Consequently, at high delithiation states, the combined effects of oxygen charge delocalization and the inherently denser packing of AB-stacked oxygen layers contribute to the contraction of interlayer spacing in the O1 phase.

In contrast, air exposure triggers uneven Li^+ extraction from particle surface, causing localized lattice strain accumulation rather than deep delithiation states. This inhomogeneous process induces partial interlayer gliding transitions from O3 to O1* stacking particularly concentrated near the surface area, as directly evidenced by SXDM and HRTEM measurements. Crucially, the limited lithium removal under chemical delithiation weakens the local shielding effect while amplifying interlayer electrostatic repulsion. These competing factors result in the expanded interlayer spacing characteristic of the O1* phase.

Moreover, our experiments indicate that the O1* phase is unstable and tends to transform into the rock salt phase under external stimuli. As shown in Figure R11, the O1* phase disappears and transforms into the rock salt phase after electron beam irradiation.

We have added a corresponding description to the revised manuscript.

Figure R10. Schematic diagram of the O3, O1, and O1* phases.

Figure R11. Microscopic TEM characterization of the SC81-air surface under beam irradiation.

12. Comment #2

Comment: *“The authors assert that surface side reactions between the surface and air in single-crystalline samples are more pronounced than in polycrystalline ones, but no polycrystalline data were provided, including electrochemical performance and characterizations, which is unconvincing. The authors should provide data on polycrystalline samples, preferably using PC samples with a similar specific area as the SC sample, to determine if the exposed surface area is the critical factor influencing performance.”*

Response: Thanks for your comments. Following your suggestion, we conducted the same electrochemical comparisons on the PC cathode to better understand the differences from the SC cathode. As presented in Figure R12. The PC-NMC811-air electrodes still show elevated charging plateaus but are much lower than those of the SC81-air samples. It is clear that the discharge capacities only have slight decreases compared to the pristine PC-NMC811 samples. Although the PC-NMC811-air electrodes present fast capacity decay, especially at 4.5 V, the capacity retentions are much better than those of the SC81-air samples. Please note that the electrochemical performance degradation of PC layered cathodes we demonstrated here is similar to the some recent publications^[6-8]. These results indicate that PC layered cathodes are less susceptible to air-induced degradation compared to SC cathodes.

SEM and TEM were employed to characterize the morphology and structure of the PC-NMC811-air sample (Figures R13 and R14). As expected, the formation of residual lithium compounds and structure degradation are confined to the outermost surface of the secondary particle, while the inner primary particles remain basically unaffected. These observations strongly support our conclusion in this work. For PC layered cathodes, despite the destruction of the outermost surface, the grain boundaries still serve as Li^+ diffusion channels for the intact internal primary particles, leading to minimal performance degradation. In contrast, for SC layered cathodes, the particle surface is the sole medium for Li^+ extraction and insertion, and its breakdown results in severe performance failure.

We would also like to clarify that we are not able to restrict the PC cathode material to have a similar specific surface area as the SC material in this comparison. The morphological differences, including variations in specific surface area, are intrinsic to PC and SC materials. It is well known that PC cathodes usually typically have a larger surface area than SC cathodes^[9]. Therefore, more surface attacks should occur on PC cathodes, yet they show less damage in electrochemical performance, which again suggests that surface reactions in SC cathodes are more detrimental.

Thank you again for your valuable feedback, which has helped us to further refine our work.

[6] Jung, R. et al. Effect of ambient storage on the degradation of Ni-rich positive electrode materials (NMC811) for Li-ion batteries. *J. Electrochem. Soc.* 165, A132 (2018).

[7] Xia, Y. et al. Industrial modification comparison of Ni-Rich cathode materials towards enhanced surface chemical stability against ambient air for advanced lithium-ion batteries. *Chem. Eng. J.* 450, 138382 (2022).

[8] Lv, C. et al. Revealing the degradation mechanism of Ni-rich cathode materials after ambient storage and related regeneration method. *J. Mater. Chem. A* 9, 3995-4006 (2021).

[9] Ni, L. et al. Challenges and strategies towards single-crystalline Ni-Rich layered cathodes. *Adv. Energy Mater.* 12, 2201510 (2022).

Figure R12. a, b, Charge/discharge curves of pristine PC-NMC811 (a) and PC-NMC811-air (b) with voltage ranges of 2.8–4.4 V at 0.1 C for first 3 formation cycles, and 0.5 C for the rest cycles. c, d, Charge/discharge curves of pristine PC-NMC811 (c) and PC-NMC811-air (d) with voltage ranges of 2.8–4.5 V at 0.1 C for first 3 formation cycles, and 0.5 C for the rest cycles. e, f, Cycle performance of pristine PC-NMC811 and PC-NMC811-air with a cutoff voltage of (e) 4.4 V and (f) 4.5 V at 0.1 C for first 3 formation cycles, and 0.5 C for the rest cycles.

Figure R13. The SEM observations of the PC-NMC811-air particles, showing the residual lithium compounds on the surface of secondary particle.

Figure R14. The SEM and STEM images of the crashed PC-NMC811-air particles. The inner primary particles show clean surfaces and no structural degradation.

13. Comment #3

Comment: “The core point that has been emphasized in this work is that “surface lattice distortions” has a greater impact on SC NCM than surface residual lithium does

on SC NCM. However, the evidence is not enough. Many characterizations in the manuscript confirmed the bad effect of surface residual lithium. It is confusing. Experimental settings need to be adjusted to prove the conclusions of the paper.”

Response: Thanks for your comment. We apologize for any confusion caused by the previous presentation of our findings. In this revision, we have thoroughly considered the impact of surface lithium residues on electrochemical performance with the addition of new results. However, through extensive characterization, we find that surface layered structure distortion, which is unrecoverable after air exposure, has a more detrimental effect on electrochemical performance. This effect is independent of the presence of surface lithium residues, as subsequent results show that electrochemical performance remains compromised, indicating that surface lattice distortions play a dominant role in degradation. We believe these revisions will provide a more comprehensive understanding of our key findings.

The effects of surface residual Li remain a topic of debate. While most studies suggest that residual Li exacerbates the cycling performance of polycrystalline cathodes, a recent study^[10] reports that coating an amorphous Li_2CO_3 layer on the surface of cathode particles can enhance both discharge capacity and cycling stability. This finding indirectly implies that the degradation of electrochemical performance is primarily driven by the disruption of the surface layered structure upon air exposure rather than the presence of residual Li itself. This perspective aligns with our conclusions, further reinforcing the notion that surface structural integrity plays a more critical role in cathode stability than previously assumed.

As shown in Figure R15, the electrochemical test results support this conclusion. The presence of Li_2CO_3 leads to an elevated charging plateau, implying increased resistance and polarization. However, the subsequent decrease in charging plateau during later cycles suggests the gradual decomposition of Li_2CO_3 . Figures R16 and R17 show 3D-CRED and particle-level EDS characterizations, confirming the complete decomposition of Li_2CO_3 after 5 cycles. These findings are clearly described in the revised manuscript to differentiate the effects of surface lithium residues on electrochemical performance. Surface lithium residues mainly play some roles on the initial electrochemical cycles. After 5 cycles of lithium residue removal, despite slight capacity recovery, the air-exposed samples still exhibit low capacity and rapid capacity decay during prolonged electrochemical cycling. This prompted us to focus on investigating the surface layered structure of the cathode.

Specifically, 3D-CRED patterns reveal the occurrence of streak diffraction in the SC81-air sample, implying the presence of stacking faults in layered structure after air storage. HRTEM observations clearly show surface structure distortions and the formation of the O1^* phase, with typical edge dislocation defects existing at the interface between the mismatched O1^* and O3 phases. GPA analysis highlights lattice strain around these defects. Additionally, synchrotron TXM and 5D-nanodiffraction data analysis further demonstrate an increase in Ni valence and surface lattice strain after air exposure. GITT testing of SC81-air after 5 cycles confirms that surface structural changes significantly hinder Li^+ ion diffusion during electrochemical processes. TEM test of the cycled SC81-air samples shows

that lattice distortions induce crack formation and extension during electrochemical cycling. These multiscale experimental results provide strong evidence for the detrimental impact of surface lattice distortions on electrochemical performance.

To further support our conclusions, we also conducted additional experiments with SC81 samples artificially mixed with Li_2CO_3 (Figure R18) and PC-NMC811-air (Figures R12-14) samples. As shown in Figures R18, Li_2CO_3 itself does not substantially affect cycling performance. Electrochemical tests on PC-NMC811 and PC-NMC811-air (Figure R12) show that PC layered cathodes are less susceptible to air-induced degradation than SC cathodes. The SEM and TEM observations reveal that the formation of residual lithium compounds and structure degradation are confined to the outermost surface of the secondary particle (Figures R13 and R14). These additional experiments strengthen our findings and further clarify the specific roles of surface lattice distortions versus surface residual lithium in determining the electrochemical performance of SC cathodes.

[10] Sheng, H. et al. An air-stable high-nickel cathode with reinforced electrochemical performance enabled by convertible amorphous Li_2CO_3 modification. *Adv. Mater.* 34, 2108947 (2022).

Figure R15. a, b, Charge/discharge curves of pristine SC81 (a) and SC81-air (b) with voltage ranges of 2.8–4.4 V at 0.1 C for first 3 formation cycles, and 0.5 C for

the rest cycles. c, d, Charge/discharge curves of pristine SC81 (c) and SC81-air (d) with voltage ranges of 2.8–4.5 V at 0.1 C for first 3 formation cycles, and 0.5 C for the rest cycles. e, f, Cycle performance of pristine SC81 and SC81-air with a cutoff voltage of (e) 4.4 V and (f) 4.5 V at 0.1 C for first 3 formation cycles, and 0.5 C for the rest cycles.

Figure R16. The 3D-CRED results of SC81-air particles before and after 5 cycles. The disordered diffraction points represent the polycrystalline nature of Li₂CO₃ surface impurity. The streak diffraction indicates the presence of lattice distortion and stacking defects in the layered structure.

Figure R17. EDS characterization of the SC81-air sample after 5 cycles.

Figure R18. The electrochemical performance plots of pristine SC81 mixed with 3.21% Li_2CO_3 .

14. Comment #4

Comment: “Some comparisons do not make much sense, such as Fig. 2d, e. First, air exposed SC81 is definitely worse than unexposed SC81. Second, surface residual lithium induced impedance increase or rock salt phase can also lead to higher capacity loss than the amount of Li ions extracted from the crystal lattice.”

Response: Thanks for your comments. We respectfully disagree that these comparisons are meaningless. Single-crystal layered oxides, as a newly morphology-designed cathode material, have not been extensively studied for their air stability. While it is true that air-exposed samples perform worse than unexposed ones, these comparisons are crucial for understanding the specific electrochemical changes induced by air exposure, particularly in terms of capacity and cyclic stability.

We agree that “surface residual lithium induced impedance increase”. But with the decomposition of Li_2CO_3 in the initial cycles, the impedance effects on capacity and charge potential gradually disappear. The increase in the charging plateau of the SC81-air sample during the first cycle indicates the decomposition of Li_2CO_3 at high voltages. The subsequent decrease in the charging plateau and capacity recovery behavior in later cycles support this statement. 3D-CRED and particle-level EDS characterizations (as shown in Figures R16 and R17) clearly confirm the complete decomposition of Li_2CO_3 after 5 cycles. Despite some degree of capacity recovery, the air-exposed samples still show rapid capacity decay after long-time cycles. GITT testing after 5 cycles indicates that the air-exposed sample continues to hinder Li^+ ion diffusion during electrochemical processes. These findings

highlight the substantial impact of surface structural changes on electrochemical performance, even after lithium residues are fully removed.

Furthermore, HRTEM characterization reveals the detailed surface structure of the SC81-air cathode after the removal of surface lithium residues. The formation of the O1* phase and induced lattice distortion are more dominant than the slight rock salt phase. We confirm that the O1* phase is unstable and tends to transform into the rock salt phase under external stimuli. This instability accelerates structural degradation towards the final rock salt phase during electrochemical cycling in air-exposed samples. Additionally, the localized lattice strain and dislocation defects formed at the O1*/O3 phase interface further exacerbate the mechanical failure of particles during electrochemical cycling.

Without these comparisons, we would not know how severe the air-induced degradation of the SC cathode is. It is not just the surface residual lithium or rock salt phase that drives degradation as conventionally believed, the lattice distortion and unstable O1* phase are critical contributors. A deeper investigation into these comparisons is essential for identifying the origin of degradation and developing further protection or improvement approaches.

In summary, these comparisons provide crucial insights into the mechanisms underlying performance degradation in air-exposed SC81 samples, emphasizing the role of surface structural changes and the instability of the O1* phase in contributing to the observed electrochemical behavior.

15. Comment #5

Comment: *“The degradation of Ni-rich cathode in air is a multi-factor process. Only several STEM or TEM images at local region is not enough to rule out other factors in the decay of SC81.”*

Response: Thanks for your comment. In this work, we conducted a comprehensive comparative study of the structure, morphology, valence states and composition of single-crystal samples after air exposure, from atomic to single-particle and electrode scale characterization, involving multi-dimensional data collection. These advanced characterization techniques include macroscopic X-ray diffraction (ex/in-situ XRD), high spatial resolution scanning X-ray diffraction microscopy (SXDM, 50 nm × 50 nm real space resolution and 0.0001 Å⁻¹ reciprocal space resolution, 5D diffraction information collection), full-field transmission X-ray microscopy (TXM, 40 nm voxel resolution), 3D-continuous rotation electron diffraction (3D diffraction data collection) and high-resolution transmission electron microscopy (HRTEM, atomistic resolution). This level of in-depth analysis has not been previously explored in such detail.

Surface residual lithium is well-known in previously reported polycrystalline systems, primarily due to its ease of detection via conventional spectral and morphological techniques. However, the changes in the surface layered structure,

beyond the traditional rock-salt phase transitions, remain insufficiently explored in the literature. This gap in understanding was one of the key motivations for our study.

Accompanying the formation of surface lithium compounds, surface structural phase transitions are subtle and challenging to detect by conventional structural characterization methods. Indeed, in our work, synchrotron-based XAS analyses only reveal slight edge shifts stemming from valence changes. As you have also pointed out, localized TEM characterization alone may not be sufficiently representative. To address this limitation and ensure a more comprehensive and convincing analysis, we employed a range of multiscale and multi-dimensional characterization techniques. In addition to localized TEM, we incorporated particle-level EDS, 3D-CRED (3D diffraction data collection), TXM tomography, and scanning X-ray nanodiffraction (5D diffraction information collection) to support our conclusions. These advanced techniques offer a more holistic view of the structural and chemical changes occurring in the cathode material during air exposure.

We believe these multiscale and multi-dimensional characterizations effectively address the complexity of degradation mechanisms of Ni-rich cathodes, providing a robust foundation for our conclusions. In addition, we appreciate the reviewer for noting our recent publication in *Science* on the multiscale characterization of single-crystal cathode materials (mentioned in the next comment), which further reflects our commitment to comprehensive analysis.

16. Comment #6

Comment: “The schematic diagram in Fig. 6g is too similar to a recently published *Science* paper Fig. 5 (<https://www.science.org/doi/10.1126/science.ado1675>). Proper revisions should be made to avoid any similarities.”

Response: Thanks for your comment. We have redrawn the schematic diagram in the revised manuscript to ensure that it is distinct from our recently published *Science* paper. The updated version of the schematic (Figure R19) highlights the key differences between polycrystalline and single-crystal layered cathodes during air exposure. Specifically, it illustrates the formation of O1* and O3 phases and their structural mismatch, which leads to structural disorder and dislocation defects. These undesirable structural changes further exacerbate crack formation and the rock salt phase transition during electrochemical cycling. We have taken care to make the revised schematic original and to clearly depict the mechanisms unique to our findings, ensuring there is no visual or conceptual similarity to previously published work.

Figure R19. New-version schematic diagram comparing polycrystalline and single-crystalline cathodes under air exposure.

17. Comment #7

Comment: “The performance and capacity are significantly inferior to previously published work of single-crystalline Ni-rich cathodes in *Nature Communications*. Such as *Nature communications*, 2021, 12(1), 5320.”

Response: Thank you for your comments regarding the electrochemical performance of our samples. The electrochemical performance of single-crystal NMC with/without modifications varies widely in literatures. We have also reviewed recent publications with similar Ni content to ours^[11-14], and the electrochemical performance of our samples is comparable to them. Electrochemical performance of cathode materials is influenced by various factors, including material synthesis, electrode preparation, and cell assembly. For example, as noted in reference 14, the shape and size of single-crystal particles can significantly impact electrochemical performance. It is important to note that the primary aim of our study is not to material optimization, but rather to investigate the degradation mechanisms induced by air exposure.

Our work focuses specifically on understanding how air exposure affects the degradation mechanisms of single-crystal Ni-rich cathodes. Through detailed characterization, we reveal surface structure changes such as lattice distortion and phase transition like O1*, which contribute to the decay in electrochemical performance. These findings provide valuable insights into developing effective surface protection strategies to mitigate the adverse effects of air exposure—a critical challenge for the practical application of Ni-rich cathodes.

We hope this response clarifies the focus of our study and underscores its contribution to advancing the understanding of air-degradation mechanisms in Ni-rich cathodes.

[11] Zhang, Q. et al. Mitigating Planar Gliding in Single-Crystal Nickel-Rich Cathodes through Multifunctional Composite Surface Engineering. *Adv. Energy Mater.* 14, 2303764 (2024).

[12] Jian, J. et al. Na⁺ orientates Jahn-Teller effect to tune Li⁺ diffusion pathway and kinetics for Single-Crystal Ni-rich LiNi_xCo_yMn_{1-x-y}O₂ cathode materials. *Chem. Eng. J.* 496, 154344 (2024).

[13] Wang, C. et al. Resolving complex intralayer transition motifs in high-Ni-content layered cathode materials for lithium-ion batteries. *Nat. Mater.* 22, 235–241 (2023).

[14] Xiao, J. et al. Assessing cathode–electrolyte interphases in batteries. *Nat. Energy* 9, 1463–1473 (2024).

Itemized Responses/Revisions to the original manuscript

Article Reference: NCOMMS-24-33182

Manuscript Title: “*Insight into structural degradation of single-crystalline nickel-rich layered cathode under air exposure*”

Responses to reviewer 1:

General comment: “*Thanks for providing the comments. All my questions are addressed, so this work can be accepted by the Journal of Nature Communications.*”

Response: Thanks for your support of our work.

Responses to reviewer 3:

General comment: “*In the revised manuscript, the author responded to my concerns in detail and convincingly. Additional electrochemical data, SEM and STEM images of PC-NCM811 cathode with/without air exposure were provided. Furthermore, 3D-CRED results of SC81-air particles before and after 5 cycles, and the electrochemical data of pristine SC81 mixed with Li_2CO_3 also addressed my concern. New data further confirmed the authors' conclusion. In my opinion, this version is now proper to be published in Nature Communications.*”

Response: Thanks for your recommendation of our work.